# A Survey on Fairness Without Demographics

**Patrik Joslin Kenfack**   *patrik-joslin.kenfack.1@ens.etsmtl.ca*
*ÉTS Montréal, Mila*

**Samira Ebrahimi Kahou**   *Samira.Ebrahimi.Kahou@gmail.com*
*University of Calgary, Mila*
*Canada CIFAR AI Chair*

**Ulrich Aïvodji**   *Ulrich.Aivodji@etsmtl.ca*
*ÉTS Montréal, Mila*

**Reviewed on OpenReview:** *https://openreview.net/forum?id=3HE4vPNIfX*

## Abstract

The issue of bias in Machine Learning (ML) models is a significant challenge for the machine learning community. Real-world biases can be embedded in the data used to train models, and prior studies have shown that ML models can learn and even amplify these biases. This can result in unfair treatment of individuals based on their inherent characteristics or sensitive attributes such as gender, race, or age. Ensuring fairness is crucial with the increasing use of ML models in high-stakes scenarios and has gained significant attention from researchers in recent years. However, the challenge of ensuring fairness becomes much greater when the assumption of full access to sensitive attributes does not hold. The settings where the hypothesis does not hold include cases where (1) only limited or noisy demographic information is available or (2) demographic information is entirely unobserved due to privacy restrictions. This survey reviews recent research efforts to enforce fairness when sensitive attributes are missing. We propose a taxonomy of existing works and, more importantly, highlight current challenges and future research directions to stimulate research in ML fairness in the setting of missing sensitive attributes.

## 1 Introduction

The growing success of Machine Learning (ML) and Deep Learning has led to successful applications in several domains such as healthcare (Shailaja et al., 2018; Jafri & Arabnia, 2009), criminal justice (Berk et al., 2019; Rudin, 2019; Berk, 2012; Tollenaar & Van der Heijden, 2013), and finance (Dixon et al., 2020; Heaton et al., 2016; Culkin & Das, 2017). The decisions provided in such domains can have profound social impacts, such as allowing or denying a loan to an individual (Pandey et al., 2017), releasing a defendant from jail (Angwin et al., 2016), admitting an individual to a university (Waters & Miikkulainen, 2014), or hiring an applicant for a certain job position (Van den Broek et al., 2021). In this regard, the use of ML systems in high-stakes decision-making has raised concerns about how these systems operate and the fairness of the decisions. Recent empirical studies have shown that a data-driven approach can unintentionally learn human biases, perpetuate or amplify them, and even introduce new ones (Caliskan et al., 2017; Bolukbasi et al., 2016; Buolamwini & Gebru, 2018; Zafar et al., 2017), leading to discriminatory outcomes against certain groups of individuals (Mehrabi et al., 2021; Barocas et al., 2017). Thus, in recent years, the social impacts of ML systems have led to an increasingly growing interest and efforts to address unfairness in machine learning.

Many definitions of fairness in ML have been proposed, along with many algorithms to achieve them (Caton & Haas, 2020). These fairness definitions and algorithms typically utilize demographic information to measure and mitigate the unfairness of ML models (Mehrabi et al., 2021; Pessach & Shmueli, 2023). They rely on the strong assumption that complete and reliable demographic information is available, which is not true in many

real-world applications. Therefore, the applicability of these fairness-enhancing algorithms is hindered by various constraints on demographic information. Specifically, in many real-world applications, demographic information is noisy (Lamy et al., 2019), partially available (Awasthi et al., 2020), and even completely missing (Lahoti et al., 2020). This has raised the need to design fairness-enhancing algorithms that can operate under various restrictions on demographic information. A naive approach in such settings consists of training the model without demographic information with the hope that the model will make predictions without relying on demographic attributes. However, protected attributes[1] could be correlated with other non-protected features that affect models' predictions (Kilbertus et al., 2017). For example, a decision-making process could not have access to the race of individuals but use their zip code. As people from the same origins tend to live in the same neighborhood, the zip code will act as a proxy for race. The model could indirectly rely on people's race to make decisions, leading to discrimination in the outcomes.

Existing surveys on algorithmic fairness generally emphasize the need for more efforts in bias mitigating without relying on demographic information (Caton & Haas, 2020; Mehrabi et al., 2021; Pessach & Shmueli, 2023; Garrido-Muñoz et al., 2021). Consequently, researchers have done intensive work to design fairness methods under different assumptions of demographic information. However, a significant limitation in the literature lies in the lack of a clear and consistent conceptualization of these assumptions, as they are expressed in different forms in existing algorithms. In addition, when demographic information is unavailable, it is often unclear what algorithm can be used to enforce each type of fairness notion, along with the underlying challenges, assumptions, and limitations. In light of this, this paper aims to provide a comprehensive survey of methods proposed to measure and mitigate unfairness in settings where the assumption of full access to reliable demographic information is not met. The assumptions on protected attributes govern the design choice of fairness-enhancing algorithms and the notion of fairness they aim to achieve. For example, when protected attributes are assumed missing, algorithms proposed to enhance fairness generally leverage information that correlates with the unknown demographic group (e.g., gradient information and errors disparity of the model) and target fairness metrics such as equal accuracy (Chakrabarti, 2023; Lahoti et al., 2020; Ahn et al., 2022). When protected attributes are assumed to be partially available, existing methods aim to enforce different group fairness notions on the data without protected attributes using techniques from semi-supervised learning (Awasthi et al., 2020; Coston et al., 2019; Kenfack et al., 2023a). On the other hand, when the protected attributes are assumed private, existing fairness methods aim to improve different group fairness notions while providing privacy guarantees for sensitive information, thereby aiming to comply with bias-free and data protection principles.

This survey identifies four primary assumptions or constraints on demographic information: (i) *missing sensitive attributes* stemming from privacy or legal restrictions (Hashimoto et al., 2018); (ii) *noisy sensitive attributes* arising from flawed data labeling or data corruption; (iii) *private sensitive attributes*, arising from constraints concerning the confidentiality of sensitive attributes (Chen et al., 2022a); and (iv) *proxy sensitive attributes*, due to the use of non-sensitive attributes correlating with the sensitive ones (Zhao et al., 2021). Or proxy attributes estimated from partial demographic (Diana et al., 2022). These constraints are not defined solely based on the reason provided but on the technical challenges resulting from the constraints on the sensitive information. Effectively addressing these constraints is crucial for advancing fairness in machine learning and ensuring the practical applicability of these methods across broader real-world scenarios. This paper proposes a taxonomy of fairness-enhancing algorithms without demographics based on their assumptions of protected attributes, their objectives, the techniques used, and the fairness notion they target. We provide a systematic review of fairness notions that do not (fully) rely on sensitive attributes and recent algorithms to achieve them.

The rest of this paper is organized as follows. We present related surveys in Section 2. In Section 3, we delve into the concept of protected attributes, exploring scenarios where the assumption of their availability does not hold. We examine how the absence of access to these attributes can give rise to unforeseen biases in the ML pipeline. Section 4 provides a background of various concepts utilized by methods presented in the survey, such as group fairness notion, fairness enhancing algorithms, and differential privacy. Fairness notions and methods to mitigate unfairness in missing sensitive attribute settings are introduced in section 5 and 6, respectively. Section 8 is dedicated to the conclusion and future perspectives.

---

[1]Throughout the paper, we use (sensitive/protected/demographic) group/attribute interchangeably

Table 1: **Existing surveys on machine learning fairness**. Most surveys do not cover or briefly (fairly) discuss fairness without demographics.

| Paper | Key topic(s) of the survey | Cover missing demographics? | Enhancements in this paper |
|---|---|---|---|
| (Ashurst & Weller, 2023) | Benefits and risks of collecting demographic data; Privacy methods for protecting demographic data; Overview of approaches for addressing missing sensitive attributes. | Yes | Fairness definitions with missing sensitive attributes; A taxonomy of approaches; Systematic review of existing methods; Current challenges and future perspectives; |
| Mehrabi et al. (2021) | Different sources of bias; type of discrimination and fairness definitions; Fair ML algorithms for classification; Fairness beyond classification; Datasets for ML fairness. | No | Sensitive attributes free fairness definitions and mitigation algorithms; Source of bias when the sensitive attribute is unknown; |
| Caton & Haas (2020) | Fairness definitions; Fairness algorithms for binary classification; Fairness beyond classification | No | Fairness definitions and mitigation algorithms without sensitive attributes; Bias audition under missing sensitive attributes. |
| Le Quy et al. (2022) | A comprehensive review of datasets set for ML research; Fairness definitions; Sensitive attributes and statistics of existing fair ML datasets | Fairly | NA |
| Zehlike et al. (2021) | Fairness definitions in ranking; Techniques to enforce fair ranking; | Fairly | NA |
| Pessach & Shmueli (2023) | Source of bias; Fairness definitions; Fair ML algorithms; Fairness beyond classification; and Datasets for ML fairness research | No | Fairness-enhancing algorithms without full access to the sensitive attributes. |
| Wan et al. (2022) | Fairness definitions; In-processing techniques for bias mitigation | Fairly | In processing techniques with unknown or noisy sensitive attributes |
| Makhlouf et al. (2021) | Causality-based fairness notions; Algorithms to estimate causal quantities | No | NA |
| Chhabra et al. (2021) | Fairness definitions for clustering; Algorithms to enforce fairness in clustering. | Yes | NA |
| Garrido-Muñoz et al. (2021) | Type of bias in NLP; Bias mitigation techniques in NLP. | Yes | NA |
| (Dunkelau & Leuschel, 2019) | Fairness definitions; Bias mitigation techniques; dataset for fair ML and fairness toolkits. | No | NA |

## 2 Related Surveys

There are several surveys within the fair ML community. Some of these surveys cover the same topics broadly but with some specificity. In this section, we briefly present some of them in position to this paper. Mehrabi et al. (2021) proposed a survey on machine learning biases, covering several sources and types of biases along with some mitigation strategies. A similar survey is proposed by Caton & Haas (2020); Wan et al. (2022); Le Quy et al. (2022); Pessach & Shmueli (2023) emphasizing bias mitigation techniques for classification tasks and beyond. Le Quy et al. (2022) survey benchmarks for machine learning research. Several datasets

Table 2: Some examples of conventions and laws against discrimination worldwide.

| LAW OR CONVENTION | GOAL |
|---|---|
| The Human Rights Act Ewing (1999) | Prevents discrimination on a wide range of grounds, including 'sex, race, color, language, religion, political or other opinions, national or social origin, association with a national minority, property, birth or any status |
| US Fair Housing Act Yinger (1999) | Protects people against discrimination for different housing services, including renting, buying, getting a mortgage, and housing assistance. The act makes unlawful any decision or action that is taken solely based on race, color, religion, or national origin. |
| CEDAW Women (1979) | The UN Convention on the Elimination of All Forms of Discrimination against Women. |
| Equal Credit Opportunity Act Hsia (1978) | Make unlawful for creditors to discriminate against applicants based on their race, color, religion, national origin, sex, marital status, or age. |
| UNESCO Convention against Discrimination in Education | Enforces anti-discrimination in education, making it available to all in any circumstances. |

are presented for various domains in machine learning, including natural language processing and computer vision. For each dataset, the set of available sensitive features is described, a causal graph between features is presented, and different fairness notions are measured. Soremekun et al. (2022); Bansal (2022) survey fairness definitions and bias mitigation technique in natural language processing. While Zehlike et al. (2021), Wang et al. (2022) and Pitoura et al. (2021) focus on fairness in ranking and recommender systems. Table 1 presents an overview of existing surveys along with the position of this work.

To complement existing surveys, this paper focuses on methods to measure and mitigate unfairness when there are various constraints over sensitive attributes, e.g., missing, limited, private, or noisy. This context has received less but growing attention within the past years, and we hope this paper will lay the groundwork for much more research effort to address fairness issues under these challenging constraints on demographic information. Most closely in the spirit of our work is the survey by Ashurst & Weller (2023). It discusses challenges and techniques to collect demographic data, methods providing protections for collection, and alternative methods such as sensitive information inference. They also presented methods for fair learning without demographic data. In contrast, this work differs on many points: We conceptualize the constraints in sensitive attributes. We identify and discuss five main constraints or assumptions over sensitive information used in existing fairness-enhancing techniques without demographic information. We present and discuss fairness notions that do not rely on sensitive information and a taxonomy of fairness-enhancing techniques without demographics based on the constraints we identified. Under each category in the taxonomy, the paper provides a systematic review of existing techniques, their pros and cons, and the fairness metric they can handle. We also highlighted the limitations, current challenges, and open research questions in bias mitigation without demographic data.

## 3 Unfairness in Automated Decision Making

This section discusses the notion of protected attributes and their importance for bias assessment and mitigation. We present various settings under which protected attributes are not fully available. We also discuss the origins of bias and their impact on ML models, especially when demographic groups are unknown.

### 3.1 Protected Attributes

Features are considered protected when they can be grounds for discrimination and their use is impertinent for decision-making. Many conventions and laws prohibit using protected attributes as a basis for decision-making. These laws and conventions seek to avoid discrimination against groups of people in various domains. Table 3.1 provides a non-exhaustive list of anti-discrimination laws and conventions worldwide. Some common protected attributes in these laws and conventions include gender, race, ethnicity, sexual orientation, religion, etc. Regulations and conventions also prohibit the collection or the use of sensitive information to ensure privacy

for individuals. These restrictions pose challenges when designing algorithms that rely on sensitive attributes to ensure fair outcomes across demographic groups.

## 3.2 Constraints on the Protected Attributes

This section covers various settings where access to complete and clean sensitive attributes is not possible, as well as their challenges for unfairness measurement and mitigation. Existing methods proposed to enforce fairness under incomplete demographic information make different assumptions about the sensitive attributes. Based on these assumptions, we identify the four main constraints on the sensitive attributes. This includes cases where sensitive attributes are entirely missing, partially available, noisy, or accessible only through related features (proxy).

### 3.2.1 Missing Sensitive Attribute

Sensitive attributes can be missing for several reasons; some of the most common reasons include constraints in the data collection process and privacy concerns.

- **Data collection**: During data collection, the sensitive attributes of people might not have been recorded. For example, users were not requested to provide their gender.

- **Privacy or legal compliance**: As people become more concerned about the privacy of their sensitive information, data-driven algorithms are increasingly subject to data protection by regulators such as the Electronic Communications Privacy Act (ECPA) and General Data Protection Regulation (GDPR). Legal compliance can restrict direct access to users' sensitive attributes.

Most existing bias assessment and mitigation methods require direct access to sensitive attributes (Dwork et al., 2012; Hardt et al., 2016). These methods are not directly applicable without sensitive attributes, making bias assessment and mitigation challenging in these scenarios. Under the missing sensitive attributes constraints, the existing algorithms aim to improve the performance of worst-performing (unknown) demographic groups. We present in Section 6.2 and Section 7 methods proposed for bias mitigation and assessment without access to sensitive attributes, respectively.

### 3.2.2 Noisy Sensitive Attributes

In some cases, sensitive attributes are available but are noisy. One of the most common reasons sensitive attributes are noisy is that they have been corrupted during data collection or estimated using a non-optimal classifier. For example, we can be interested in assessing the fairness of a facial recognition system across different demographic groups. However, demographic information is not included in many image datasets. One may rely on a different classifier trained to predict demographic attributes to obtain the group labels, which are likely noisy (Buolamwini & Gebru, 2018). Additionally, sensitive attributes are likely to be corrupted when users must self-report their sensitive information (Rosenman et al., 2011).

### 3.2.3 Private Sensitive Attributes

A common reason for missing demographic information is restrictions from laws or regulations that prohibit collecting and using sensitive information about individuals in algorithmic decision-making. For instance, the GDPR prevents the use of racial information about customers. At the same time, these laws and restrictions enforce non-discrimination in automated decision-making systems. On the other hand, automated decision-making systems require protected attributes to audit and mitigate discrimination from the system. This raises the new challenge of design methods that comply with these two seemingly contradicting principles, i.e., design methods to build fair models while preserving the privacy of sensitive information. To alleviate the privacy restrictions, sensitive attributes can be made available under privacy-preserving mechanisms, i.e., a mechanism operating on the data that can expose or use sensitive data with strong privacy guarantees for individuals' demographic information (Dwork et al., 2014). We further discuss this in Section 6.4.

### 3.2.4 Proxy Sensitive Attritutes

In some cases, sensitive information is not mandatory during the data collection, and reluctant users might prefer not to provide their sensitive information for privacy reasons. For example, credit card companies can collect personal information to assess creditworthiness. While some information is mandatory for risk assessment, others, such as demographic information, might be optional. As a result, only a few data points will have a value for the sensitive attributes. In such a scenario, data imputation approaches such as replacing the most frequent value or inferring the missing values from other features could be used (Coston et al., 2019) to obtain pseudo labels (proxy) for the missing sensitive attributes. However, this should be done carefully as incorrect estimation of the demographic information can lead to more harm (Cf. Section 6.1.2). In other settings, the sensitive attributes are available in a different but related dataset or task (Awasthi et al., 2021). For example, in a loan application task where the feature gender is missing, a related task could be a model that predicts gender based on other information in a separate context. Although the feature gender is not directly utilized in the loan application model, the insights gained from predicting gender in the related task might indirectly inform decision-making processes, contributing to bias mitigation without explicitly using gender on the base task (Buolamwini & Gebru, 2018). On the other hand, non-sensitive information that correlates with unknown sensitive information might exist. These related attributes (proxies) can also control fairness w.r.t the unknown sensitive attributes (Zhao et al., 2021).

From the decision-maker viewpoint, these constraints on the sensitive attributes seem conceptually overlapping. More specifically, decision-makers might not have access to sensitive information in private and missing demographic setups. However, from a technical perspective, these constraints represent the core assumptions that govern the design of the algorithms used to mitigate unfairness in each setting. For example, methods under private attribute setups generally assume some access to sensitive attributes and involve some privacy-preserving mechanisms. On the other hand, in the missing sensitive attribute setup, algorithms are designed to improve the performance of unknown minority groups implicitly and without privacy preservation concerns. Therefore, the constraints on sensitive information are distinguished by technical contributions and challenges in enforcing fairness under each constraint.

## 3.3 Discrimination in Machine Learning

Unfairness or discrimination in a decision-making process made by humans is quite clear: it occurs when the outcome of a decision systematically depends on an individual's protected attribute and not on characteristics that are useful in assessing that individual's abilities with respect to the task or desired outcome. For example, an officer assessing a consumer's loan application may be qualified sexist or racist if the decision to refuse the loan is based on the applicant's sex or race. Not relying on non-sensitive features, such as the customer's financial or production management capabilities, to repay the loan can be considered discriminatory. Moreover, even when decision-making processes do not directly use demographic information, there might be rules (proxy to demographic information) leading to a disparate impact on some demographic groups. Machine learning models are good at learning patterns related to unobserved sensitive information.

Models are trained using historical data, and the goal is to discover patterns (general trends) and make predictions about future data. This training process, therefore, does not aim to discriminate against individuals based on their group membership (unless the model is trained for that purpose). However, unfairness in ML systems is more similar to *systemic discrimination* that also exists in some human decision-making processes (Craig, 2007). Systemic discrimination happens when the decision-making process is, often unintentionally, less or more advantageous for some groups of people. For example, a hiring decision process that considers applicants' criminal records may have disparate outcomes for equally qualified applicants (despite a race-neutral hiring rule) because of racial disparities in criminal records caused by discrimination in policing (Bohren et al., 2022). Therefore, direct discrimination in policing leads to systemic discrimination in the hiring process. Specifically, systemic discrimination can be the result of past successive direct discrimination.

Performances of ML systems are generally evaluated using different metrics such as accuracy, correct classification rates, misclassification rates, or error w.r.t a given loss function. The fairness of these models is often assessed in terms of disparities in performances across different demographic groups. As a result,

fairness in machine learning is a subjective notion that can be defined based on three main aspects: the metric used, the type of task, and the learning paradigm. For example:

- **Classification**: The model is considered unfair when the accuracy is higher for one group than another (Barocas et al., 2019). Unfairness is also measured by comparing the correct classification and/or misclassification rates (Zafar et al., 2017; Hardt et al., 2016). More generally, fairness is measured using metrics derived from the confusion matrix of the classifiers. The ratio or difference between considered metrics for each protected subgroup can be used to measure the disparities.

- **Generative models**: For models such as generative adversarial networks (GANs) Goodfellow et al. (2020)–where two competing networks are used to estimate the true distribution of the data–the model is considered unfair when the generator at the test time samples data from one protected subgroup more often than another (Kenfack et al., 2021a). Tan et al. (2020) and Kenfack et al. (2022) measure unfairness in generative models using the Kullback-Leibler (KL)-divergence between the distribution of protected subgroups and the uniform distribution. Hence, a generative model is considered fair when the distribution of subgroups in the generated data is uniform, i.e., all demographic groups are equally represented.

- **Reinforcement learning (RL)**: Chi et al. (2021) defined fairness in RL as parity in the reward returned for different demographic groups. Unfairness occurs when the returned reward of two policies trained of different groups sharing the same states and action spaces is higher for one group than another.

- **Natural language processing**: the model is considered unfair or biased when it encodes social biases such as racial or gender stereotypes from the data. For example, in machine translation, translation from gender-neutral languages such as the Turkish language to non-gender-neutral languages such as English tends to assign articles such as "she" to professions such as nurses and housekeepers. While assigning articles "he" to professions such as Doctor, Engineer (Prates et al., 2020).

- **Raking system**: The model is considered unfair when it under-ranks individuals from protected groups (Zehlike et al., 2021). Fairness is measured using a ranking score function that measures the disparities disparity between demographic groups in the top-$k$ ranking results. A ranking result gets a lower score if the top-ranked results consist of samples from mostly one group.

As can be seen, fairness in various areas of machine learning involves analyzing and quantifying the impact of trained models on different demographic groups. Therefore, addressing unfairness in machine learning systems involves identifying the source of bias, quantifying the performance disparities in different demography groups, and developing methods to mitigate them.

## 4 Background

This section provides an overview of the concept used by various methods covered in the paper. The background information provided in this section represents the basic building blocks for existing techniques proposed for bias mitigation when complete and clean sensitive attributes are unavailable, as presented in Section 6. We present different source biases in the absence of the sensitive attributes, group fairness metrics, fairness enhancing methods, and differential privacy used by some work to build fair models with privacy guarantees on the sensitive attributes. A reader familiar with these concepts can skip this section.

### 4.1 Group Fairness Metrics

In general, group fairness metrics are defined by quantifying the disparities of metrics that can be derived from the confusion matrix of each demographic group; see Makhlouf et al. (2021) for an exhaustive list of group fairness metrics. The most popular group fairness notions include:

| Notation | Description |
|----------|-------------|
| $X$ | Random variable defining data sample. |
| $Y$ | Random variable defining the class label |
| $A$ | Random variable defining the protected attribute. |
| $\hat{Y}$ | Random variable defining the predicted class. |
| $\hat{A}$ | Random variable describing corrupted or predicted sensitive attributes. |
| $a_i$ | Sensitive attribute of the sample $i$ |
| $\hat{a}_i$ | Predicted or corrupted sensitive value of sample $i$ |
| $x_i$ | data sample sample $i$, $x_i \in \mathbb{R}^d$, described with $d$ features. |
| $y_i$ | Label of sample $i$ |
| $\hat{y}_i$ | Predicted label of sample $i$ |

Table 3: Table of Notation

- *Statistical Parity* (SP): also known as *demographic parity*, this fairness notion requires that the classifier positive outcome must be independent of the protected attributes (Dwork et al., 2012), i.e., $A \perp \hat{Y}$. A classifier achieves statistical parity if the following expression is satisfied

$$P(\hat{Y} = 1 | A = 0) = P(\hat{Y} = 1 | A = 1) \tag{1}$$

However, a classifier that satisfies this notion of fairness could also lead to discrimination against individuals from the non-protected group. For example, consider a model for hiring and gender as the sensitive attribute. If the goal is to hire the ten most qualified candidates and the hiring process uses statistical parity as the fairness notion, the hired candidates should have equal gender representation. Therefore, if the ten most qualified candidates are men, the model will select five less qualified women to achieve statistical parity, harming five candidates from the non-protected group (violating *individual fairness*). Moreover, when the sensitive attribute is correlated with the class label, a classifier that achieves statistical parity cannot provide perfect predictions (Verma & Rubin, 2018).

Statistical parity can also be measured using the ratio between the fractions of disadvantaged and advantaged samples assigned to the positive class. The ratio is generally referred to as *Disparate Impact* (DI) and measured as follows:

$$DI = \frac{P(\hat{Y} = 1 | A = 0)}{P(\hat{Y} = 1 | A = 1)} > 1 - \epsilon \tag{2}$$

Typically, one sets $\epsilon \approx 0.2$, which suggests $DI > 0.8$ for a fair classifier, as stated by the four-fifths rule of maximum acceptable disparate impact proposed by the US Equal Employment Opportunity Commission (EEOC) (Zafar et al., 2017).

- *Equalized Odds* (EOD) (Zafar et al., 2017; Hardt et al., 2016): This fairness notion promotes the conditional independence between the classifier outcome and the sensitive attribute given class label, i.e., $A \perp \hat{Y} | Y$. Thus, Equalized Odds is based on the confusion matrix and promotes the true positive rates (TPR) and the false positive rates (FPR) across groups. Specifically, a model satisfies equalized odds if

$$P(\hat{Y} = 1 | Y = y, A = 0) = P(\hat{Y} = 1 | Y = y, A = 1)$$
$$y \in \{0, 1\} \tag{3}$$

In our example, the hiring model will satisfy equalized odds if it gives all groups the same advantages (TPR) and disadvantages (FPR). That is, candidates labeled as qualified or unqualified should have a similar classification rate, regardless of gender.

- *Equal Opportunity* (EOP) (Hardt et al., 2016; Zafar et al., 2017): In some applications, one might be more interested in being fair when a positive outcome is made. Equal Opportunity is similar to

equalized odds but focuses on equal TPR across groups and thus promotes *equal true positive rate* across groups. Specifically, a model will achieve equal opportunity if

$$P(\hat{Y} = 1|Y = 1, A = 0) = P(\hat{Y} = 1|Y = 1, A = 1) \tag{4}$$

Similarly, one might be concerned about fairness when the model gives a positive outcome while the true one is negative. The fairness notion to promote in this scenario will be *equal false positive rates* across groups, also known as equal False Discovery Rate (Zafar et al., 2017). We refer the readers to Verma & Rubin (2018) and Mehrabi et al. (2021) for a more comprehensive list of group fairness notions.

- *Overall Equal Accuracy (Berk et al., 2021)*: This notion of fairness requires all demographic groups to have similar accuracy. This notion could be important in applications where the quality of service is crucial for users since individuals experiencing low-quality service might refrain from using the system. For example, Buolamwini & Gebru (2018) evaluated three commercial gender classifiers by Microsoft, IBM, and Face++. Their evaluation demonstrated that females with darker skin tones receive the worst accuracy across all classifiers, while lighter skin gets the best accuracy. Specifically, a model achieves equal accuracy if :

$$P(Y = \hat{Y}|A = 0) = P(Y = \hat{Y}|A = 1) \tag{5}$$

For our example, the hiring model will satisfy equal accuracy if all groups receive similar accuracy. However, this metric does not guarantee equalized odds since the TPR and FPR might differ between groups, even though groups have equal accuracy (Angwin et al., 2016).

## 4.2  Fairness Enhancing Methods

Existing fairness-enhancing algorithms can be grouped into three main approaches: pre-processing (Madras et al., 2018; Kamiran & Calders, 2012; Kenfack et al., 2023b), in-processing (Kamishima et al., 2011; Bechavod & Ligett, 2017; Noriega-Campero et al., 2019), and post-processing (Hardt et al., 2016; Nabi & Shpitser, 2018). This depends on whether the notion of fairness is enforced before the model training, during the model training, or after training the model. This section covers fairness-enhancing techniques —that rely on sensitive attributes— leveraged by some exciting works presented in Section 6.1 for bias mitigation when sensitive attributes are noisy or predicted. The method covered here involves post-processing (Hardt et al., 2016) and in-processing techniques (Zhang et al., 2018; Agarwal et al., 2018).

**Calibration (Hardt et al., 2016).**  Fairness-enhancing post-processing techniques involve treating the model as a black box and enforcing fairness constraints by adjusting the model's output. Hardt et al. (2016) proposed an optimization problem over the model's output $(\widehat{Y})$ to derive a classifier $(\widetilde{Y})$ that satisfies fairness constraints (Equalized Odds) while minimizing the classification loss. When the model output is continuous (a score function), the derived classifier is based on a threshold of each demographic group to maximize the classification loss while satisfying fairness constraints, i.e., equal opportunity and equalized odds. Similar methods are proposed in the literature, and they differ mainly in how the optimization problem is defined.

**Adversarial Debiasing.**  Proposed by Zhang et al. (2018), this adversarial-based approach enforces the independence between the classifier outcome and the sensitive attributes. The classifier's output is used as input for the adversary network, which tries to predict the sensitive attribute. In the minimax optimization problem, the goal of the classifier is to prevent the adversary from predicting the sensitive attribute, thus enforcing statistical parity. The adversary considers the predicted outcome and ground truth to enforce equalized odds. Thus, fooling the adversary enforces the independence between the model's output and the sensitive attributes. To enforce Equal Opportunity, the adversary gets the classifier outputs only for samples with positive outcomes.

**Exponentiated gradient.**  The exponentiated gradient is a reduction approach for fairness that transforms a classification problem with fairness constraints into a sequence of cost-sensitive classification problems.

The new problem is solved by the Exponential Gradient method that looks for the saddle point where the classification loss is minimized and fairness is maximized (the disparities are minimal). The method yields randomized classifiers for which classifiers with the lowest prediction error satisfying the fairness constraint are returned (Agarwal et al., 2018). This approach can work for various base models (e.g., Logistic regression and Random Forest) and with most existing group fairness metrics.

### 4.3 Differential Privacy (DP)

Differential Privacy (DP) (Dwork et al., 2006) is used by some existing works presented in Section 6.4 to address fairness concerns under private sensitive attribute's constraints. DP is a mathematical framework to formally quantify the privacy/utility trade-offs of an algorithm that operates on personal data. It seeks to maintain the privacy of individuals when releasing results derived from a confidential database by limiting the impact of any single record on the outcome and incorporating noise. Differential privacy finds applications across various domains, but its significance in machine learning is especially notable. In machine learning, differential privacy helps in training models on private data (Abadi et al., 2016) while ensuring that the trained model does not memorize or leak specifics about the training data, thus preventing potential attacks such as membership inference (Shokri et al., 2017). Differential privacy can be applied in a centralized manner (Global DP) when data subjects trust a data analyst to enforce privacy when realizing the result of an algorithm that requires their data, using different mechanisms such as the Laplace mechanism (Dwork et al., 2014) and the Exponential mechanism (McSherry & Talwar, 2007). It can also be applied in the absence of a trusted centralized party (Local DP), allowing data subjects to directly create noisy versions of their private inputs through protocols like randomized response (Warner, 1965), initially designed to address evasive answer bias in social science, allowing useful analysis of responses provided for embarrassing or sensitive questions. More formally, DP for the sensitive attribute can defined as follows:

**Definition 1 (Differential Privacy)** *Given $\epsilon \geq 0$, $\delta \in [0,1]$. A randomized mechanism $\mathcal{M}$ is $(\epsilon,\delta)-$differentially private if, for any adjacent datasets $D$ and $D'$, i.e, datasets that only differ with a single entry, we have*

$$\mathrm{P}(\mathcal{M}(D) \in R) \leq \exp(\epsilon) \cdot \mathrm{P}(\mathcal{M}(D') \in R) + \delta \tag{6}$$

*Where $R \in \mathcal{R}$ is a subset of the output response, $\epsilon > 0$ the privacy budget with values close to 0 meaning strong privacy, and $\delta$ the probability of the algorithm not being $\epsilon$-DP.*

Abadi et al. (2016) proposed DP-Stochastic Gradient Descent (DP-SGD), a modification of the gradient descent algorithm which provides provable privacy guarantees. DP-SGD bounds the sensitivity of sample gradients and adds noise to the gradient before updating the weight models. DP-SGD provides privacy guarantees for deep learning models trained on datasets containing sensitive information.

## 5 Fairness Notions without Sensitive Attributes

In most existing group fairness notions, individuals are grouped based on their sensitive attributes to achieve fairness across these groups. Thus, classical group fairness notions are not directly applicable when the sensitive attributes are unobserved. This section discusses fairness definitions that do not rely on sensitive information for measurement. Figure 1 showcases an overview of existing fairness notions that do not assume access to sensitive attributes.

### 5.1 Individual Fairness

Proposed by Dwork et al. (2012), this definition requires that similar individuals with respect to a given task should receive similar outcomes. Within this fairness notion, the sensitive attribute used to group individuals is irrelevant as fairness is assessed at the individual level, making individual fairness feasible even when the demographic information is unavailable as long as the distance metric between individuals is given and is irrelevant to sensitive information.

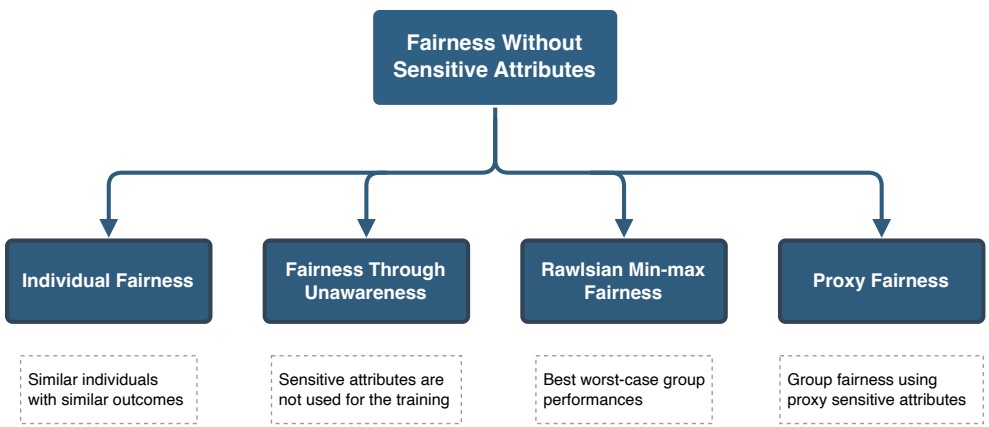

Figure 1: Overview of fairness notions without sensitive attributes.

**Definition 2 (Individual Fairness (Dwork et al., 2012))** *Given a distance metric on individuals: $d : V \times V \to \mathbb{R}$, a mapping function from individuals to the outcomes probability: $M : x \to \Delta(x)$; $x \in V$, and a distance metric $D$ over the distribution of outcomes. $M$ achieve individual fairness iff:*

$$D(M(x), M(y)) \leq d(x, y). \tag{7}$$

Here, $(D, d)$ are fixed distance metrics, and $x, y \in V$ are individuals. Individual fairness properly captures the $(D, d)$-Lipschitz property and provides meaningful fairness guarantees. Dwork et al. (2012) showed that when the classifier satisfies the Lipschitz property, it also achieves statistical parity with a certain amount of bias. For example, the model for scoring resumes will achieve individual fairness if applicants with similar resumes get similar predictions. Suppose two applicants $x$ and $y$ are very similar: they have similar years of experience, graduated from the same school, and have mastered the same programming languages. However, $y$ is a woman; the similarity metric should ignore this fact since gender is irrelevant to determining who should be hired. Thus, the distance between $x$ and $y$ should be small, e.g., $d(x, y) = .01$. If the model $M$ assigns the probabilities .9 and .7 to $x$ and $y$ respectively, the distance between their scores $D(M(x), M(y)) = .2$, assuming $D$ is defined as the statical distance (Dwork et al., 2012; Fleisher, 2021), which is higher than the distance between individuals. Therefore, $M$ fails to satisfy individual fairness since the *Lipschitz mapping* in 2 is not satisfied, suggesting similar applicants are not treated similarly.

Despite its fairness guarantees, individual fairness suffers from the assumption that task-specific similarity metric between individuals is given, which is hard to measure in practice (Lahoti et al., 2019; Dwork et al., 2012).

Relaxations of individual fairness have been proposed for various purposes, such as solving the problem related to the task-specific similarity metric or generalizing individual fairness from a training set to the underlying population. In this regard, Yona & Rothblum (2018) proposed *approximate metric-fairness*, a relaxation of individual fairness with guarantees of generalization on the underlying population. This notion of fairness allows for a small fairness error. It requires that, for two individuals sampled from the underlying population, with all but a small probability, if they are similar, they should be treated similarly (Yona & Rothblum, 2018). The statistical distance between the classification distributions is used to measure the similarity between two given individuals. Thus, a model $h$ is said to be $(\alpha, \gamma)$-*approximately metric-fair* w.r.t. the metric $d$ and the distribution $\mathcal{D}$ if

$$\Pr_{x, x' \sim \mathcal{D}} \left[ \left| M(x) - M(x') \right| > d(x, x') + \gamma \right] \leq \alpha \tag{8}$$

Where $d$ represents the similarity metric, $\mathcal{D}$ the data distribution, $\gamma$ is the small additive slack in the similarity measure, and $\alpha$ the fraction of pairs of individuals for which the metric-fairness does not hold. However, this notion also assumes that the task-specific distance metric between individuals is given. Although this fairness notion opens the door to fairness-generalization bounds, there exist settings where the fairness error it accepts might harm certain individuals in the population.

## 5.2 Fairness Through Unawareness

This definition of fairness used in some contexts, such as in law enforcement in certain countries, requires that sensitive attributes should not be explicitly used in the decision system (Kusner et al., 2017). This means the model is considered fair as long as the model does not rely on sensitive attributes during training or decision-making. For example, in the case of resume screening, the model is considered fair as long as the candidates' resume does not disclose information about their gender, and the model does not explicitly use gender to make its predictions. However, it's important to note that the model being unaware of the sensitive attributes is insufficient to avoid discrimination in the outputs.

While not using sensitive attributes directly in the decision system is a step toward fairness, it is not a foolproof solution. For instance, certain features can be correlated with sensitive attributes, allowing the model to rely on them implicitly. For example, research has shown that the age at which someone starts programming correlates with gender (Barocas et al., 2019). Similarly, factors such as salary expectations and working hours per week may tend to be lower for female applicants, or the zip code can be correlated with the origins of some applicants as people from the same demography tend to live in the same neighborhood. While these proxy features can be relevant to the decision, they may reflect existing biases and lead to unfair decisions. For instance, how long someone has been programming is a factor that gives us valuable information about their suitability for a programming job. Still, it also reflects the reality of gender stereotyping (Barocas et al., 2019). Thus, while this fairness notion seeks to prevent the explicit use of sensitive attributes, it is important to know that information about these attributes can still leak into the model through other means.

## 5.3 Rawlsian Mini-Max Fairness

The Rawlsian Mini-Max fairness notion does not rely on sensitive attributes but on a notion of unknown least advantaged groups, which decision-makers should define. It is derived from social sciences and is based on the principle of distributive justice proposed by John Rawls (Rawls, 1999). Rawl's principle of justice states (among other things):

> "*Social and economic inequalities are to be arranged [...] to the greatest benefit of the least advantaged members of society, consistent with the just savings principle.*" (Rawls, 1999, p. 226)

In other words, this principle suggests that the right decision maximizes the minimum outcome, i.e., the decision that makes the worst outcome as good as possible (Rawls, 2001). More formally, Rawlsian fairness can be defined in the context of machine learning as follows:

**Definition 3 (Rawlsian Max-Min Fairness)** *Given a set of hypotheses $\mathcal{H}$ and unknown demographic groups $a \in A$, a hypothesis $h^* \in \mathcal{H}$ achieves Rawlsian Max-Min fairness if it maximizes the accuracy of the worst-off groups, i.e.,*

$$h^* = \arg \max_{h \in \mathcal{H}} \min_{a \in A} U_{\mathcal{D}_a}(h) \tag{9}$$

where $U_{\mathcal{D}_a}$ is the expected utility/accuracy of the hypothesis $h$ over the group $a$. The group information $(a)$ is unknown in the general setting. The intuition is that making the worst case as good as possible would positively impact truly disadvantaged groups.

While this notion has gained attention in recent years as a potential solution to achieving fairness without sensitive attributes, it also inherits the critiques of the Rawlsian notion of distributive justice. Among these criticisms is the emphasis on the notion of the least advantaged, which may not consider demographics or individuals within the population (Altham, 1973). The least advantaged is not a single individual but a

group that is difficult to define. Rawls proposed to select least advantaged groups from the least fortunate with respect to (i) family and class, (ii) natural endowments, and (iii) fortune and luck (Rawls, 1999). In the context of algorithmic fairness, particularly for classification, the least advantaged group can be formed from mostly misclassified samples. Therefore, a model that aims to achieve Rawls's principle by focusing on improving misclassified samples, or samples from the worst-off distribution in a set of distributions created from the empirical distribution, might not only fail to improve fairness with respect to the truly disadvantaged group but may also make the model highly sensitive to outliers (Hashimoto et al., 2018; Lahoti et al., 2020). In contrast to other group-based fairness notions, Rawlsian fairness does not aim to mitigate the disparities in a given metric across groups but to make the worst-performing group as good as possible. The Rawlsian definition can also effectively handle intersectionality between groups and be applied beyond classification tasks as long as the utility metric is defined. Its application is particularly pertinent in medical applications (Ricci Lara et al., 2022) or any other domains where decreasing the model performance to achieve parity is unacceptable.

### 5.4 Proxy Fairness

Proxy fairness notions are derived from group fairness notions. They assume correlated features are known and can be used as *proxy* to measure or enforce fairness with respect to the true protected attributes.

Considering scenarios where direct access to sensitive attributes is impossible, *proxy fairness* notions rely on features that correlate with the sensitive counterparts. These proxy-sensitive attributes can also be predicted through an attribute classifier. The underlying idea is to utilize correlated or predicted features as *proxy* for the actual sensitive attributes. In particular, the sensitive attributes for group fairness metrics presented in Section 4.1 can be replaced by the proxy-sensitive attributes to measure fairness in the model. By evaluating the selected group fairness notion with respect to these proxies, one can measure the fairness of models w.r.t *true* sensitive attributes. However, a drawback of proxy fairness notions lies in their assumption that the features correlating with unobserved sensitive features are known, and determining the degree of correlation proves challenging without input from a domain expert. Furthermore, auditing model fairness based on proxy might lead to underestimating or overestimating the actual fairness violation. Special care must be taken in the design of proxy-sensitive attributes for optimal bias estimation (we provide further discussion in Section 7).

## 6 Fairness Enhancing Techniques Without Demographics.

In the previous section, we presented fairness notions that do not (fully) rely on sensitive attributes. In this section, we review methods proposed to enforce based on the constraints we identified in Section 3.1, i.e., sensitive attributes are predicted, missing, noisy, or private.

### 6.1 Enforcing Fairness Using Proxy Sensitive Attributes.

Proxy-sensitive attributes can be obtained using related non-sensitive features or using limited demographic information available. Proxy attributes must be designed carefully, as incorrect proxies or predictions of sensitive attributes may result in adverse effects, such as sub-optimal solutions or incorrect disparities estimation in the model (*cf.* Section 7). In this section, we review methods to enforce fairness using proxy-sensitive attributes. Table 4 showcases an overview of existing methods, the datasets, and fairness metrics used for evaluation. Each method is also grouped based on mechanism type, i.e., at which step of the ML pipeline the fairness mechanism can be applied: at the data level (pre-processing), during the model training (in-processing), or after training the model (post-processing); Existing methods are also grouped based on proxy features obtained via related features or partial demographic information.

#### 6.1.1 Using Related Features.

In some applications, non-sensitive features may be closely associated with sensitive ones (Zhao et al., 2021; Gupta et al., 2018; Diana et al., 2022). Proxy features are non-sensitive features that correlate with the sensitive ones. For example, (Elliott et al., 2009) showed that customers' origins can be estimated using

Table 4: **Overview of methods to enforce fairness using proxy sensitive information.** Proxy-sensitive information is obtained either using demographic predictors or related features. Existing methods in this category can improve group fairness metrics (e.g., equalized odds, equal opportunity, and statistical parity) w.r.t to the true sensitive attributes.

| Papers | Related Features | Partial Demographic | Mechanism Type | Evaluation Metrics | Datasets | General Description |
|---|---|---|---|---|---|---|
| FairRF (Zhao et al., 2021) | ✓ | ✗ | In | • Accuracy 
 • EOP 
 • SP | • Adult 
 • Compas 
 • LSAC | Assumes the knowledge of non-sensitive features correlating with unknown sensitive ones and adds a regularization term to minimize their influence over the classifier's output. |
| NOCCO (Pelegrina et al., 2023) | ✓ | ✗ | Pre | • Accuracy 
 • EOP | • Adult 
 • Compas 
 • LSAC 
 • Taiwanese credit | Proposes a preprocessing method to detect sensitive features in the datasets using the Hilbert automatically–Schmidt independence between class labels and features splitting the data into groups. |
| KSMOTE (Yan et al., 2020) | ✓ | ✗ | Pre | • F1-score 
 • EOP 
 • EOD | • Adult 
 • Compas 
 • Violent crime | The proposed method relies on clustering techniques to restore class balance while improving model fairness on downstream tasks without needing to observe sensitive attributes. |
| Proxy-learner (Diana et al., 2022) | ✗ | ✓ | In | • Accuracy 
 • EOP 
 • EOD | • FolkTable datasets | The work shows that training a model to predict the sensitive attributes can be a good substitute for the ground truth sensitive attributes when the latter is missing. |
| CGL (Jung et al., 2022) | ✗ | ✓ | Pre | • Accuracy 
 • EOP | • Compas 
 • CelebA 
 • UTKFace | The paper proposes a confidence-based attribute classifier that randomly assigns group labels on samples with low confidence prediction and shows its benefits on fairness in downstream tasks. |
| SR-CVAE (Grari et al., 2021) | ✓ | ✗ | In | • Accuracy 
 • EOP 
 • EOD | • Adult 
 • Default Data | Assuming a causal graph of the data, the approach uses Bayesian inference to reconstruct the sensitive attribute in the latent space and uses an adversarial approach to enforce fairness. |
| FairDSR (Kenfack et al., 2023a) | ✗ | ✓ | Pre | • Accuracy 
 • EOP 
 • EOD 
 • SP | • Adult 
 • LSAC 
 • Compas 
 • CelebA | The paper proposes an uncertainty-aware sensitive attribute classifier to improve fairness-accuracy tradeoffs on downstream tasks with missing sensitive attributes. |
| MTL-fair (Aguirre & Dredze, 2023) | ✗ | ✓ | In | • Accuracy 
 • EOP | • Clinical-Notes 
 • Online-Reviews 
 • Twitter | A multi-task learning setup where the partial demographic is available in one task, and the objective is to improve fairness on tasks with missing sensitive attributes. |
| FairDA (Liang et al., 2023) | ✗ | ✓ | In | • Accuracy 
 • F1-score 
 • EOP 
 • SP | • Toxicity 
 • Compas 
 • Adult 
 • CelebA | A domain adaptation setup where partial demographic information is available in one domain. An adversary is used to enforce fairness in the target domain where the demographic information is missing. |
| Proxy-fairness(Gupta et al., 2018) | ✓ | ✗ | Post | • Accuracy 
 • EOP | • Adult 
 • Default Data | Study bias mitigation when proxy-sensitive attributes are used. Shows that proxy-sensitive attributes can improve fairness, but they can also overestimate or underestimate it. |
| BiFair (Ozdayi et al., 2021) | ✗ | ✓ | In | • Accuracy 
 • EOP 
 • EOD 
 • SP | • Adult 
 • Bank data | Proposes a bilevel optimization where the partial demographic is used to compute sample weights, and the target classifier is trained with a weighted loss to improve fairness on samples with missing sensitive data. |
| FURL-PS (Zhang et al., 2022) | ✗ | ✓ | Pre | • Accuracy 
 • EOD | • CelebA 
 • UTKFace | Uses the partial demographic information to train a contrastive sample generator, which generates images with edited sensitive attributes. A contrastive loss is adapted to improve fairness in the latent representation. |
| APOD (Wang et al., 2023) | ✗ | ✓ | Pre | • Accuracy 
 • EOD 
 • EOP | • Loan default 
 • Adult 
 • German 
 • MEPS | An active learning setup where the goal is to select the optimal data points with missing sensitive attributes, for which human annotations will improve fairness for the protected groups. |

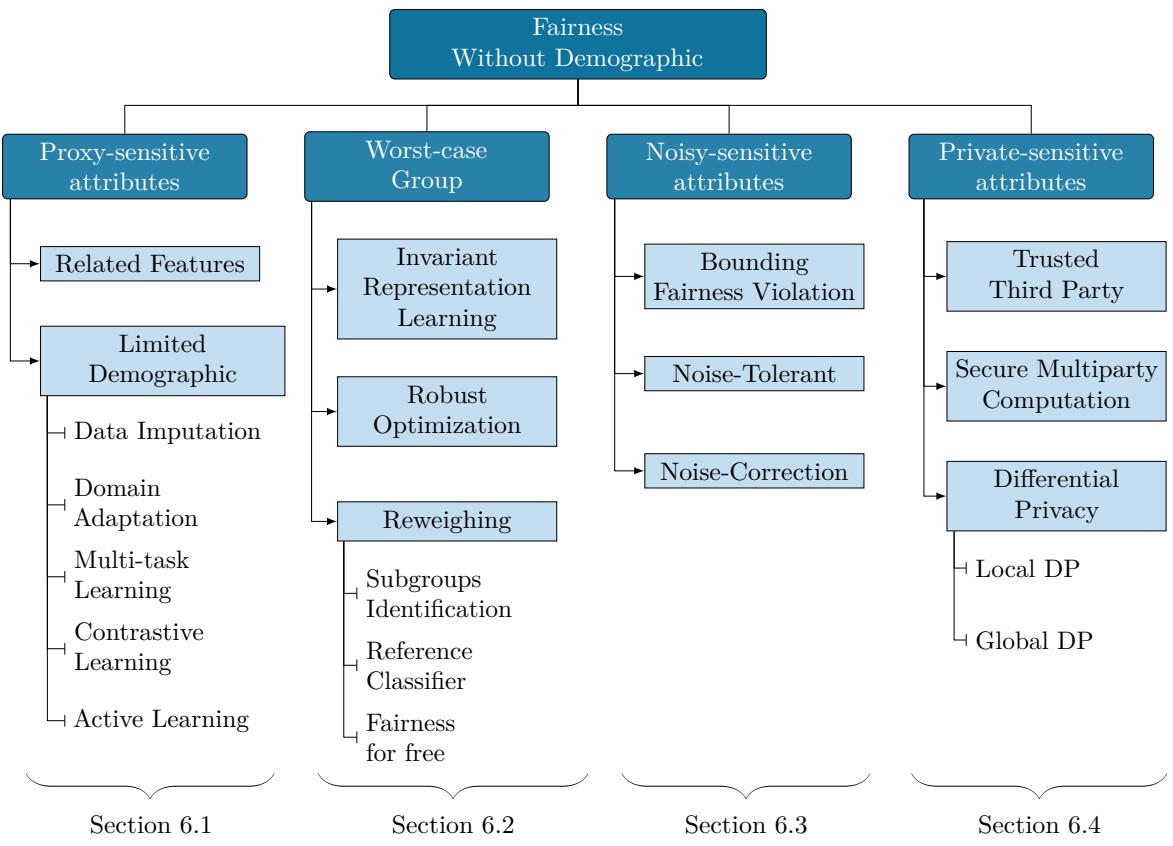

Figure 2: A taxonomy of fairness enhancing techniques without demographic information

their last name and residence address. Also, in the Adult Census Income dataset Asuncion & Newman (2007), the protected feature *Gender* is correlated with non-protected features such as *Relationship*, *Age*, and *Marital-Status* (Le Quy et al., 2022; Zhao et al., 2021). Intuitively, if we reduce the correlation between proxy features and the model's prediction, we indirectly reduce the correlation with the unknown sensitive attributes. A straightforward approach is to train the model without the proxy features. The problem with such approach is that, since proxy features can also be useful for the target task, we would inevitably observe a drop in accuracy. Besides, it is important to design methods that control the fairness-accuracy tradeoffs. Using the post-processing fairness technique by Hardt et al. (2016) (cf. Section 4), Gupta et al. (2018) demonstrated empirically that enforcing fairness constraints with the proxy features can improve fairness for the *true* protected groups. The authors characterize *good proxy* as features that share the same semantics with the true demographic group. However, poor proxies can hurt accuracy and fairness, especially when the proxies do not correlate equally with the true unknown sensitive attributes. Domain experts could provide the weight of the relationship between features. However, obtaining accurate values can be difficult in some real-world applications. To overcome this, Zhao et al. (2021) proposed FairRF, a framework that can learn the weights of related features during the optimization process. The framework includes a regularization term to minimize the correlation, $\mathcal{R}(x_j, \hat{y})$, between the classifier's output ($\hat{y}$) and each related feature ($x_j$). The optimization problem of the framework is defined as follows:

$$
\begin{aligned}
\min_{\theta,\lambda} \quad & \mathcal{L}_{cls} + \eta \cdot \sum_{j=1}^{K} \lambda_j \cdot \mathcal{R}\left(\mathbf{x}_j, \hat{\mathbf{y}}\right) + \beta\|\lambda\|_2^2 \\
\text{s.t.} \quad & \lambda_j \geq 0, \forall f_j \in \mathcal{F}_S; \quad \sum_{j=1}^{K} \lambda_j = 1
\end{aligned}
\tag{10}
$$

where $\mathcal{L}_{cls}$ is the classifier loss (typically a cross-entropy loss), $\eta$ controls the regularization term, $\{x_j\}_{j=1}^{K}$ is the set of highly correlated features, and $\{\lambda_j\}_{j=1}^{K}$ weights associated to each related feature. To avoid trivial

solutions, weights are regularized ($\|\lambda\|_2^2$) to enforce non-zero weights for each related features. Equation 10 is solved through alternating direction optimization (Goldstein et al., 2014), wherein the parameters $\theta$ and $\lambda$ undergo iterative updates, with one variable fixed while the other is updated in each iteration.

However, Equation 10 requires a strong assumption on feature dependencies and can result in a sub-optimal correlation between related features and the protected attribute. In other words, the true highest related features of the sensitive attribute might be hidden, making it difficult for humans to identify them correctly. More specifically, Equation 10 does not consider data-specific and distribution similarity for identifying the highest related features. To alleviate this issue, Chang et al. (2023) propose a framework to automatically identify highly correlated non-sensitive features and mitigate their influence in the model during the training. The method involves a sensitive attribute reconstruction model leveraged by a self-attention mechanism (Vaswani et al., 2017) to learn the interactions (weights) between the sensitive and non-sensitive features. The detected biased feature interactions are then mitigated via a regularization term to mitigate their influence. The method, however, requires demographic information at the training time to uncover the correlated features. In a more principled way, Pelegrina et al. (2023) proposed a statistical method to automatically detect the sensitive features without relying on real sensitive ones. The proposed method is formalized around the hypothesis that a given feature might be considered sensitive if it splits data into two groups of people (e.g., gender male or female) and the Hilbert–Schmidt (Gretton et al., 2005) independence between this feature and the target label is high. Put differently, when a feature divides individuals into groups and exerts significant influence over the target variable, it is highly probable that disparities will emerge. For example, the authors show that by applying the proposed method to the Adult dataset, the features *Marital-Status* and *Relationship* provide a high Hilbert–Schmidt score and are thus returned as sensitive as sensitive features. These features are known to correlate with gender in the Adult dataset, suggesting the efficiency of the method in uncovering sensitive features when they are unknown automatically. The method can used as a preprocessing step that produces sensitive features toward which fairness will be enforced or measured on the downstream task.

A common source of unfairness is the under-representation of the protected group in the dataset (Mehrabi et al., 2021). One way to mitigate it is to balance the representation of each group in the dataset (Kamiran & Calders, 2012). Leveraging this observation, Yan et al. (2020) addressed representation bias by adapting the K-Means SMOTE algirthm (Last et al., 2017) for *fair class balancing*. K-Means SMOTE works in three steps. In the first step, clusters are created using any clustering algorithm. In the second step, samples closest to the cluster boundaries are removed since these samples are more likely to be misclassified. Finally, in the last steps, for each cluster, the least represented classes are considered minorities, from which new samples are generated (i.e., oversampled) based on the nearest neighbors within the minority class. Although this sampling technique does not rely on sensitive features, the authors showed that it can improve fairness (w.r.t statistical parity and equal opportunity) in downstream tasks. This line of work shows the effectiveness of clustering methods in identifying groups even when their sensitive features are unknown. However, such methods are prone to bias in approximating the sensitive groups, as they require a strong assumption that clusters correlate with sensitive attributes. Alternatively, Grari et al. (2021) assumes a causal graph underlying the training data and uses the causal dependencies between variables to reconstruct the sensitive attribute (proxy) using a variational autoencoder (VAE) (Kingma et al., 2019), under a Gaussian prior for the sensitive attribute. VAE is used to model exogenous variables in causal graphs. Within the causal graph, the variables are grouped into two subsets: the subset of variables not caused by the sensitive attribute and those caused by sensitive and non-sensitive features. The first step of the framework leverages the variables and the class label to learn a latent space that contains as much information about the sensitive attribute as possible. The encoder and the decoder are implemented as neural networks optimized to obtain a latent space that reconstructs the sensitive attributes. A fair classifier is then built using an adversarial debaising approach to enforce the independence of the predictions and the learned latent representation (proxy demographic information), which is obtained from the learned posterior approximated by the encoder.

### 6.1.2 Using Limited Demographic Information.

Also referred to as *demographic scarce regime*, this setting occurs when the sensitive attributes are not explicitly collected for the target task dataset (Bharti et al., 2022; Kenfack et al., 2023a; Coston et al., 2019)

or are they are available in different a dataset. Specifically, this setting generally consists of two datasets $D_1$ and $D_2$, described by $\{X, Y\}$ and $\{X, A\}$, respectively. $D_1$ contains the class label and is used to train the target task, while $D_2$ contains the demographic information. While some works assume that $D_1$ and $D_2$ share the same feature space, others assume a distribution shift between the datasets. In this setting, one is interested in enforcing fairness on $D_1$ using demographic information in $D_2$ or estimating bias of a model trained on $D_1$ using demographic information available in $D_2$. In the literature, most methods proposed to address fairness concerns in the demographic scarce regime are based on established machine learning algorithms or paradigms used in semi-supervised learning (Van Engelen & Hoos, 2020). These adaptations include techniques such as data imputation, domain adaptation, multi-task learning, contrastive learning, and active learning.

**Data imputation.** In this line of work, $D_2$ is used to train a sensitive attributes predictor. The attribute predictor is then used to augment $D_1$ such that it jointly observes the non-sensitive feature, the class label, and the demographic information (proxy), i.e., $\hat{D}_1 = \{X, Y, \hat{A}\}$, where $\hat{A} = \{\hat{a}_i\}_{i=1}^{N}$ is the proxy sensitive attribute obtained from the attribute predictor. A highly accurate sensitive attribute predictor can help better quantify or mitigate the unfairness of models built from the augmented dataset $\hat{D}_1$. The core challenge here is to characterize the properties of good attribute predictors, as incorrect or noisy predictions may result in adverse effects, such as incorrect disparities estimation in the model or the amplification of disparities. We cover in section 7 techniques to assess bias using the proxy-sensitive attributes.

In this regard, Diana et al. (2022) and Ozdayi et al. (2021) demonstrate that if the dataset with sensitive attributes ($D_2$) is distributed identically to the data without sensitive attributes ($D_1$), a demographic predictor, known as the proxy model, can accurately estimate the missing sensitive features and fairness enforced on $\hat{A}$ can improve fairness for the true protected groups. The results suggest that practitioners accessing only a limited demographic information can impute the missing ones and still improve fairness for the true sensitive attributes. However, the performances achieved are not as good as what could be attained by using true sensitive attributes. To overcome this limitation, Kenfack et al. (2023a) designed a framework for efficient uncertainty estimation of the sensitive attribute. Under this framework, they demonstrated that enforcing fairness constraints primarily on samples with lower predictive uncertainty can enhance fairness-accuracy trade-offs. Likewise, Jung et al. (2022) use a confidence threshold over the predictions of the proxy model to discern instances where sensitive attributes are predicted with low confidence. Subsequently, random attribute values are assigned to samples with sensitive attributes predicted low confidence. These values are drawn from the empirical conditional distribution over the sensitive attributes given the class label. The authors posit that this random labeling serves as a form of regularization. In the same spirit, Diana et al. (2022) show that if the proxy model $\hat{A}$ is multi-accurate over different groups, then a downstream task with fairness constraints w.r.t $\hat{A}$ can achieve a similar level of fairness as it would have with the true sensitive features.

**Domain adaptation.** Another line of work formulates fairness in the limited demographic setup as *domain adaptation* problem (Schumann et al., 2019; Madras et al., 2018; Coston et al., 2019; Liang et al., 2023). In domain adaption, the goal is to train a model on a source domain to perform well on a target domain, with a constraint that both domains might be drawn from different but related distributions, e.g., robustness to distribution shift. See (Farahani et al., 2021) for a review on domain adaptation.

Formulated as a fairness problem, the objective is to build a fair model in a domain where the sensitive attribute is available and then transfer the fairness properties to the domain where the sensitive attribute is missing. For example, Coston et al. (2019) assume that limited demographic information is available in a source or a target domain. When demographic information is available in the source domain, the authors propose to learn the weights of each sample such that they are as close as possible to the covariate shift weights[2] subject to prevalence constraints for fairness. The covariate shift weights are measured as $q_X(x)/p_X(x)$, where $p_X(x)$ is the density distribution of data from the source domain and $q_X(x)$ the density of the target domain. The prevalence constraint ensures that all pairs of groups are close to each other. When demographic information is available in the target domain, the authors propose to learn sample weights by minimizing a double loss consisting of the fairness loss (group disparity) and the classification loss. The group

---

[2]Loss function minimizes the $L_1$ between the predicted weights and the covariate shift weights

disparity loss is then defined as the difference between the classifier score across demographic groups and is measured on the target domain in which demographic information is available. Liang et al. (2023) propose a dual adversarial approach to build a fair model on a target domain where the sensitive attribute is solely accessible in the source domain. In the source domain, a sensitive attribute is trained with a domain adversary to enforce domain invariance in the latent space. Meanwhile, in the target domain, the label classifier is trained alongside an adversary, preventing the prediction of pseudo-sensitive attributes of samples. These pseudo-sensitive attributes are derived from the attribute classifier trained on the source domain. The general objective of these methods is to transfer the knowledge of the sensitive attribute from one domain to improve fairness in the domain with missing sensitive attributes. In the same spirit, Madras et al. (2018) proposes a transferable fair representation learning approach. The transferability property of the fair representation can be particularly useful when the sensitive attribute is missing on the downstream task.

**Multi-task learning (MTL).** Aguirre & Dredze (2023) formalized the problem in a multi-task learning framework. In MTL, the goal is to optimize different but related tasks simultaneously during the training. The model is generally designed by defining a shared encoder and task-specific layers for each task (Zhang & Yang, 2018). The idea is that the knowledge gained from learning one task can benefit the performance of other related tasks. To handle missing sensitive attributes, Aguirre & Dredze (2023) considered two tasks. The first task involves training a classifier to predict the class label using the dataset with missing sensitive attributes ($D_1$). The other task consists of training the classifier with fairness constraints on the dataset with sensitive attributes. The intuition is that fairness constraints on the second task will improve fairness on the task with missing sensitive attributes. The authors also proposed a generalized approach to intersectional fairness, where the two tasks are trained with fairness w.r.t different sensitive attributes, e.g., the first task can have *Gender* as the sensitive attribute, and the second task *Race*. The proposed methods show improvement in fairness w.r.t the missing sensitive attribute across each task.

**Contrastive Learning.** As an unsupervised approach, contrastive learning aims to construct a latent space where samples sharing similar features are drawn together while distancing those that differ (Chen et al., 2020). To obtain a representation that exhibits this property, positive examples consist of paired views of the same image, whereas negative examples involve pairs of views from different images. The loss function incorporates a distance metric between examples, intending to minimize the distance between positive pairs and maximize the distance between negative pairs. Park et al. (2022b) show that contrastive learning can rely on sensitive attributes to better optimize the contrastive loss, i.e., in the latent space, the separation between dissimilar samples is more effective for some demographic groups than others. As a result, the learned representation will incur disparate performance on downstream tasks. The authors integrate a group-wise normalization factor to contrastive loss to improve the fairness of the representation. However, the proposed fair contrastive loss fully depends on demographic information. To address this, Zhang et al. (2022) propose a contrastive learning approach to acquire fair representations despite having limited demographic information. The available demographic data is utilized to train an image attributes editor. Upon receiving an input image, the editor produces its contrastive counterpart, i.e., an image with a different sensitive attribute while preserving other visual features. For instance, it can generate a realistic photo resembling a man when presented with an image of a woman. The process involves a sensitive attribute classifier for labeling the generated contrastive examples. Images with confidently labeled attributes are then used to enhance the attribute editor in a mutually improvement manner iteratively. This first step aims to generate an augmented and balanced dataset where each sample has a contrastive example with a different sensitive attribute. In the second step of the method, a fair representation is acquired through contrastive learning applied to the augmented dataset. To obtain a fair representation, the authors propose considering contrastive samples with different sensitive attributes as positive examples and negative samples as views originating from different images with the same sensitive attributes. Feature weighting is integrated into the contrastive loss, assigning higher importance to specific features that depend on the sensitive attribute in the latent space. This method, however, heavily depends on the attribute editor model, and its application is limited to image datasets. Chai & Wang (2022) propose a contrastive learning approach to learn fair representations and use limited labels to guide the training. A weighting scheme is used in contrastive loss to up-weight samples with higher classification errors on the validation set. Using the weighted contrastive loss, the representation learned is

enforced to ensure high separability of samples from minority groups, thus improving fairness on downstream tasks in terms of equalized odds.

**Active learning**    Active learning is a prevalent machine learning method in semi-supervised learning that addresses scenarios with costly data labeling. The model is initially trained on a small proportion of labeled data and predicts labels for unlabeled data Settles (2009). Instances predicted with low confidence are then referred to an oracle for annotation, e.g., a human annotator. The newly annotated data points are then used to enhance the model in the next training round. The objective is to reduce the number of queries for annotators. To learn a fair model, one can request demographic information of certain samples but with compensation. The objective is to train a fair model with a minimum number of *demographic query* (Liu & Lan, 2020).

In this spirit, Wang et al. (2023) propose an active learning approach for developing fair models under limited demographic information. The goal is to find the optimal instance for sensitive attribute annotation that can help improve the model's fairness in the next iteration while minimizing the number of label queries. In each iteration, the model is trained to minimize the classification loss with regularization over the labeled data points. The regularization term penalizes the model for any disparities, such as equalized odds or equal opportunity. In the subsequent step, a sample from the unlabelled set is selected to promote bias mitigation maximally. The sample selection uses the demographic group of samples with the highest classification error. Groups of samples are defined using predicted sensitive attributes provided by an attribute classifier. The sample selected for annotation is at a maximum distance with annotated samples in the latent space.

On the other hand, Liu & Lan (2020) explore the decoupled fair model proposed by Dwork et al. (2012), where a distinct model is trained for each demographic group. The authors introduce two strategies for selecting data points to query demographic information: 1) using samples with the most significant outcome variations across each model and 2) using samples that, when added to the labeled dataset, amplify the disparity of the current model. While empirical results show the efficacy of these query strategies, the method necessitates retraining the model after each demography query, which can be computationally expensive.

Based on demographic predictors, existing methods require the assumption that non-sensitive characteristics are highly descriptive of sensitive characteristics. However, in some applications, this assumption may not always be true. On the other, predicting sensitive information poses ethical concerns and, in some cases, is unlawful. Kenfack et al. (2023a) suggest that obtaining datasets with high uncertainty in the inference of the sensitive attribute can yield models (trained without fairness constraints) that inherently exhibit fairness properties. Additionally, Ozdayi et al. (2021) introduce BiFair, an approach structured around a bilevel optimization setup. This optimization process uses the limited demographic information to compute the samples' weights instead of trying to reconstruct the missing sensitive attributes with a demographic predictor. In the bilevel optimization process, the inner optimization minimizes the classification loss. In contrast, the outer optimization computes the fairness loss for samples with sensitive attributes and leverages these computations to determine sample weights such that they minimize the fairness loss.

### 6.1.3 Evaluation Protocol in Proxy Demographic Setting.

When fairness constraints are enforced over features correlating with the sensitive features, the evaluation process does not differ greatly from the setting where demographic information is observed. The particularity is that sensitive attributes are not directly used during the training phase, but the correlated features Zhao et al. (2021), the causal graph Grari et al. (2021), or the clusters Yan et al. (2020) are considered. For example, Zhao et al. (2021) use *Age*, *Relationship*, and *Marital-Status* in the Adult dataset as the features correlating with the unknown sensitive feature: considered *Gender*. For the Compas dataset, the unknown sensitive attribute is *Race*, and its correlated features are *Score*, *Decile Text*, and *Sex*. The evaluation of the fairness performance follows most existing works where the sensitive attributes are known at test time, and fairness performances are reported using the true sensitive attributes.

In settings where partial demographic information is available, most approaches generally split the dataset into three subsets: A subset where the sensitive attributes are missing, a subset as the dataset where the partial demographic is available, and a subset for the evaluation as the test set where the sensitive attribute

is also available. For example, Kenfack et al. (2023a), Diana et al. (2022), and Ozdayi et al. (2021) split the Adult dataset into 60%, 20%, 20% for the training, labeled demographic, and test dataset respectively. As an ablation study, some methods (Zhang et al., 2022; Ozdayi et al., 2021; Wang et al., 2023; Jung et al., 2022) generally vary the ratio of the dataset with labeled sensitive attributes to see its impact on the proposed method.

For evaluation, methods using proxy demographic information generally improve group fairness metrics, such as equalized odds, equal opportunity, and statistical parity. This is possible thanks to proxy demographic information that helps control the true demographic groups.

## 6.2 Enforcing Rawlsian Max-Min Fairness

The main objective of the Rawlsian Max-Min Fairness notion is to minimize the model error on the least advantaged protected groups. When group information is available during the training, some methods are proposed to optimize for the group performing worst (Martinez et al. (2020); Diana et al. (2021); Sagawa et al. (2019)). However, defining the least advantaged groups becomes challenging when the necessary criteria are not provided. Nevertheless, the Rawlsian fairness notion can be particularly useful to address the major challenge of intersectional fairness, where the model performance could be worse across an intersection of sensitive attributes, e.g., gender & race, age & nationality. For example, Buolamwini & Gebru (2018) showed that most commercial face recognition systems are significantly less accurate in identifying darker-skinned females. Pursuing Rawlsian fairness or improving model performance for the worst-case group shares a common goal with Out-Of-Distribution (OOD) generalization. OOD aims to enhance the model's performance on an (unknown) test domain that differs from the training domain and is not independent and identically distributed (i.i.d) (Ben-Tal et al., 2009; Shen et al., 2021; Arjovsky et al., 2019; Krueger et al., 2021). To achieve fairness, domains are defined as demographic groups to learn a model that generalizes effectively across different groups[3]. The optimization objective, in this case, is to minimize the error of the domain/group with maximum error:

$$\mathcal{R}_{\mathcal{F}}(\theta) = \max_{e \in \mathcal{F}} \mathcal{R}_e(\theta) \tag{11}$$

A model ($\theta$) that does not control the worst-case loss across groups has a low overall loss, i.e., $\mathcal{R}(\theta)$, but a high worst-case loss ($\mathcal{R}_{\mathcal{F}}(\theta)$). We group methods to improve model performance on the unknown worst-case groups into three categories: robust optimization, reweighting, and invariant representation learning. Table 5 shows an overview of existing methods, the datasets, and fairness metrics used for evaluation.

### 6.2.1 Robust Optimization

When demographic information is known a priori, Sagawa et al. (2019) proposed Group Distributionally Robust Optimization (DRO) to improve model performance on groups with higher loss. Unlike the classical Empirical Risk Minimization (ERM), DRO does not optimize for the average loss. When demographic information is not observed, Hashimoto et al. (2018) showed that EMR does not control the worst-off groups, and the authors adapted DRO (Duchi et al., 2016) to achieve Rawlsian Max-Min Fairnes by minimizing the error of the worst-case distribution of perturbations around the empirical distribution, i.e.,

$$\mathcal{R}_{\mathrm{dro}}(\theta; r) = \sup_{\mathcal{Q}:\, D(\mathcal{Q}\|\mathcal{P}) \leq r} \mathbb{E}_{\mathcal{Q}}\left[ l(\theta; X) \right] \tag{12}$$

Where $D\left(\mathcal{Q} \| \mathcal{P}\right)$ is the $\mathcal{X}^2$-divergence between the empirical distribution ($\mathcal{P}$) and probability distributions around $\mathcal{P}$ at a distance $r$ and $l(\theta; X)$ represents the model loss over samples $Z$. The authors showed that $\mathcal{R}_{\mathrm{dro}}(\theta; r)$ upper bounds the error of unknown worst-case groups, and optimizing it can help to control the model performance over the worst-case group. In practice, the problem 12 can be solved by using the following dual:

---

[3]In general, the terms "domains," "environments," and "groups/subgroups" can be used interchangeably

Table 5: **Overview of methods to enforce Rawlsian Max-Min fairness.** Methods under this category use robust optimization or a reference classifier to identify unknown subgroups. They generally improve fairness in terms of worst-case group performance (*Group Acc/Err*). Some methods can also improve Equalized Odds (EOD) and Disparate Impact (DI).

| Papers | Subgroup Identification | Robust Optimization | Reference Classifier | Mechanism Type | Evaluation Metrics | Datasets | General Description |
|---|---|---|---|---|---|---|---|
| Yenamandra et al. (2023) | ✓ | ✗ | ✓ | Pre | • Precision | • Waterbirds
• CelebA
• NICO++ | Use a "weak" model to amplify bias, then uncover samples from worst-case groups by clustering the latent space. |
| (Chai et al., 2022) | ✗ | ✗ | ✓ | In | • Accuracy
• EOD
• DI | • New Adult
• Compas
• CelebA | Upweight data points by training the model with soft labels from knowledge distillation. |
| Hashimoto et al. (2018) | ✗ | ✓ | ✗ | In | • Group Acc.
• EOP | • Twitter | Distributionally Robust Optimization to control the utility of the worst-case groups. |
| Chakrabarti (2023) | ✓ | ✗ | ✓ | In | • Group Acc. | • Adult
• Compas
• LSAC | Upweight samples identified with statistical significance as members of the worst-case group. |
| Idrissi et al. (2022) | ✗ | ✗ | ✗ | Pre | • Group Acc. | • Waterbirds
• CelebA
• MultiNLI
• Civil-Comments | Data subsampling or reweighing can achieve state-of-art performance on worst-case groups. |
| CVaR (Duchi et al., 2019) | ✗ | ✓ | ✗ | In | • Group Acc. | • MNIST
• ImageNet | Optimizes for worst-case distribution around uncertainty set of the empirical distribution. |
| JTT (Liu et al., 2021) | ✓ | ✗ | ✓ | In | • Group Acc. | • CelebA
• Waterbirds
• MultiNLI
• Civil-Comments | Use a reference classifier to find misclassified samples to be up-weighted to improve model performance on the worst groups (unknown). |
| ARL Lahoti et al. (2020) | ✓ | ✗ | ✗ | In | • Group AUC
• EOP | • Adult
• Compas
• LSAC | Weighted empirical risk minimization with adversarial reweighting |
| BPF Martinez et al. (2021) | ✗ | ✓ | ✗ | In | • Group Err. | • Adult
• Compas
• LSAC
• MIMIC-II | Optimize worst-case groups with minimum harm on best-case groups under Pareto optimality. |
| Ahn et al. (2022) | ✓ | ✗ | ✓ | In | • Group Err. | • CMNIST
• MB-MNIST
• CelebA
• Civil-Comments | Reweight data points proportionally to their gradients to improve the performance of the worst-case groups. |
| LfF Nam et al. (2020) | ✓ | ✗ | ✓ | In | • Group Acc. | • CMNIST
• CIFAR-10 | It uses a generalized cross entropy to uncover worst-case group samples and then upweight them. |
| EIIL Creager et al. (2021) | ✓ | ✗ | ✓ | In | • Group Acc. | • CMNIST
• Adult
• Waterbirds
• Civil-Comments | Train a reference model using soft-group assignment to identify worst-case group samples. |
| Ko et al. (2023) | ✗ | ✗ | ✗ | In | • Group Acc. | • CIFAR100
• Tiny-Imagenet | Show the utility of DNN ensembles for improving worst-case group performance. |

$$\min_{\theta \in \Theta} \mathbb{E}_{\mathcal{Q}} \left[ l(\theta; X) - \eta \right]^2 \tag{13}$$

Where $\eta$ is a hyperparameter controlling the strength of the robust loss. By minimizing the equation 13, the model exhibits improvement across all directions around the data-generating distribution, leading to reasonable performance across sufficiently large subgroups, particularly for the worst-case unknown groups Hashimoto et al. (2018). Hu & Chen (2022) built upon Hashimoto et al. (2018) to improve fairness in Survival Analysis, where the goal is to "*model the amount of time that will elapse before a critical event of interest happens*" such as death or patient recovery. Soma et al. (2022) propose a generalized DRO with faster convergence and tighter bound over the worst-off groups. While Peet-Pare et al. (2022) argued that enforcing fairness on models with static distributions does not capture the dynamic environment in which these models are deployed. Consequently, such an approach may not effectively model most real-world scenarios with fairness concerns. They showed that in repeated risk minimization, while DRO and ERM have similar convergence behavior, ERM converges to a fixed point where the model is biased towards majority groups. DRO converges to a fair fixed point. Peet-Pare et al. (2022) adapt *performative prediction* to distributional robust optimization and studied its properties for fairness. Performative prediction (Perdomo et al., 2020) studies a more dynamic setup, wherein the predictions of the model influence the data distributions on which the model makes subsequent predictions.

It is worth noting that DRO operates as a reweighing approach, wherein, unlike ERM, which assigns equal weights to all data points, DRO up-weights samples where the model incurs a high loss. While the robust optimization process has demonstrated its benefit to the (unknown) worst-case group, its key drawback is its sensitivity to outliers, i.e., the model could focus on optimizing noisy outliers rather than the real worst-case protected groups. To overcome the limitation of DRO, Martinez et al. (2021) propose Blind Pareto Fairness (BPF), an algorithm that can improve the worst-case performances of all groups of a certain size while ensuring the solution is also Pareto-optimal, i.e., ensures that there is no other solution that provides better group risks uniformly for all groups. With Pareto optimality, the method optimizes for the worst-case groups with a minimum performance drop for the best-performing groups.

Papadaki et al. (2022) use robust optimization to enforce fairness in a *federated learning* setup when demographic information is unknown. In federated learning, the dataset is not centralized but distantly available on different devices. Each device trains a model using its local data and sends its weights to the central model that will average all weights from the clients. The core advantage of federated learning is to enhance individuals' privacy as their data are not stored on a centralized server but locally locally (Li et al., 2019). Still, federated learning does not prevent data stored locally from leaking since model parameters or gradient updates can reveal sensitive information (Fredrikson et al., 2015). In their work, Papadaki et al. (2022) use a robust loss in each client to improve the worst-case performance of all groups of a certain size in the centralized model.

### 6.2.2 Reweighing

Reweighing techniques typically operate by devising a mechanism to identify subgroups where the model exhibits worse performance. The approach involves assigning greater weights to samples from these subgroups, thereby enhancing the model's performance on minority groups. An alternative group of methods aims to show the benefit of certain established deep learning techniques on fairness. These techniques include knowledge distillation (Chai et al., 2022), ensemble learning (Ko et al., 2023; Kenfack et al., 2021b), and class balancing (Idrissi et al., 2022). We delve into both weighing using *subgroup identification* and implicit weighing with techniques that improve *fairness by design*.

**Subgroups identification.** To overcome the sensitivity of DRO to noise, Lahoti et al. (2020) propose Adversarially Reweighted Learning (ARL) that leverages the notion of *computationally identifiable groups*. These groups are regions where the model makes more mistakes. The method uses an adversarial approach where an adversary learns samples' weights by maximizing the learner (classifier) error while the learner minimizes the classification weighted loss function. Intuitively, the adversary will assign higher weights to

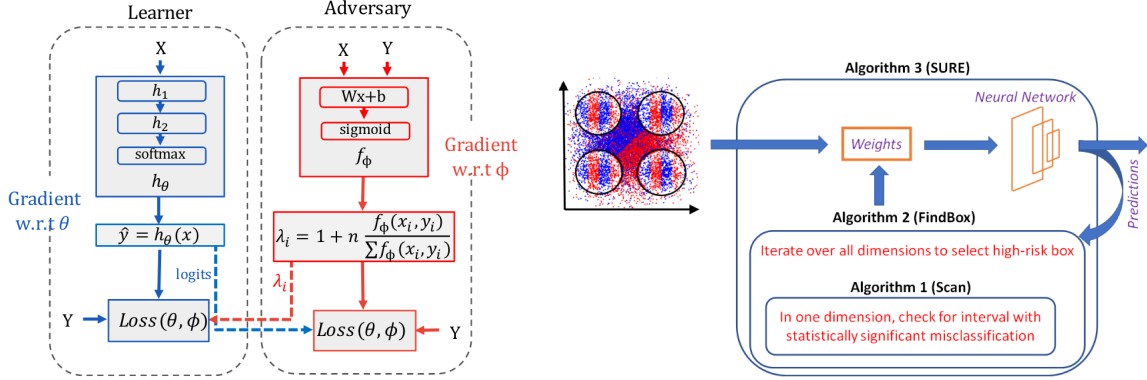

Figure 3: Overview of Adversarially Reweighted Learning (ARL) (Lahoti et al., 2020)

Figure 4: Overview of Significant Unfairness Risk Elimination (SURE) (Chakrabarti, 2023)

samples where the classifier makes the most misclassification, and to minimize its error, the learner must focus on improving on these samples. Figure 3 shows an overview of ARL.

Chakrabarti (2023) argues that ARL may encounter challenges in comprehensively identifying all subgroups due to the variability of unfairness patterns across training steps and datasets. The author suggests a strategy to address this challenge: selecting the region in the latent space where the model exhibits *statistically significant* errors and upweighting samples within that chosen region even for the correctly classified samples for the next training iteration. The emphasis on the statistical significance of unfairness risk, specifically the higher misclassification rate within the selected region, proves valuable in averting the upweighting of misclassified samples resulting from randomness during training epochs. In cases where no region is identified with a significant risk of unfairness, the training in the next iteration reverts to the classical EMR without sample weights. To identify subgroups, the latent feature vectors for correctly and incorrectly misclassified samples in the current iteration are segmented into bins (Cf. Figure 4). The number of samples per bin is a hyperparameter. The subgroup with a high unfairness risk comprises samples belonging to the bin with the most significant misclassification error across all latent feature vectors. To ensure that the selected subgroup is large enough and handles the intersectionality of groups, the selected bin is merged with samples from the surrounding bins. Empirical results show evidence that identifying subgroups with statistically significant worst-case error can outperform ARL in various ranges of settings (Chakrabarti, 2023).

**Leveraging a Reference Classifier.** Another line of work leverages the information an auxiliary model (reference classifier) provides to identify samples from disadvantaged groups. Existing methods empirically show that an auxiliary model's errors or gradient can be used to identify unknown worst-case groups (Ahn et al., 2022; Zhao et al., 2023). Fairness is then enforced using a reweighting scheme based on subgroups identified by the auxiliary model, hoping that the new weight will improve the performances of true protected groups.

Nam et al. (2020) empirically demonstrated that a model can easily fit groups satisfying spurious correlations in the data (*bias-aligned* samples) at the early stage of the training, while groups that do not satisfy spurious correlation (bias conflicting samples) are fitted later. For example, in the Waterbird dataset, waterbirds with a water background and landbirds with a land background are bias-aligned samples. In contrast, landbirds with a water background are *bias-conflicting* samples. Following this observation, the authors proposed a debiasing scheme that consists of two networks trained simultaneously. The "biased" network is fitted to amplify its early-stage predictions using generalized cross-entropy loss. At the same time, the unbiased model is enforced to focus on the mistakes of the biased model. As the biased model mostly learns the spurious correlations, misclassified samples are mostly bias-conflicting samples that are upweighted to mitigate the bias. The approach demonstrated its efficacy in improving the model's performance over the wort-case group. Building upon these results, Liu et al. (2021) introduce "Just Train Twice" (JTT), a two-step framework

designed to enhance the model's performance on worst-case groups without relying on group information during the training. In the first step, a non-complex model is trained on the target task for a few steps, intentionally leading the model to fit bias-aligned samples better, thus underfitting bias-conflicting samples. The set of misclassified samples by the reference classifier represents a proxy for the worst-case groups. In the second step, the misclassified samples, which mostly contain data points from the worst-case group, are up-weighted to enable the model to improve on this group of samples. The ability of the reference classifier to effectively learn spurious correlation depends on the number of steps used to fit the classifier, which is a hyperparameter. The authors perform upweighting by oversampling misclassified by a factor $\lambda_{\text{up}}$, which is also a hyperparameter. As a drawback, JTT requires expensive hyperparameter tuning and, most of the time, requires sensitive attributes in the validation set to achieve good results. To address this limitation, Veldanda et al. (2023) proposes a mechanism to generate pseudo-sensitive attributes efficient for tuning hyperparameters over the training set. In the same spirit, Yenamandra et al.'s (2023) approach involves leveraging a *weak model* to uncover bias-conflicting samples in the data. The authors demonstrate that bias can be amplified using a weak model, trained to deliberately underfit the training set, which maximizes the separation between minority and majority groups. In the subsequent step, samples from minority groups (bias-conflicting samples) are derived by clustering the bias-amplified latent space obtained in the initial phase, combined with the embedding of visual concepts of each image to preserve the semantic coherence of the clusters (Radford et al., 2021). These clusters are formed by fitting a Gaussian mixture for each class and sorting them based on cluster accuracy in ascending order. The top $k$ clusters are then returned as samples representing minority groups, which can be upweighted to mitigate bias in the downstream step. Methods using a single reference classifier, however, suffer from their sensitivity to hyperparameters. This sensitivity does not provide a reliable way to identify and compute the weights of samples from minority groups. In particular, for a pair reference classifiers trained with different random initialization or learning rates, the number of samples from the unknown protected group identified by each model changes significantly (Kim et al., 2022). To address this, Tiwari et al. (2024) train the reference classifier using earlier layers of the model, which is more effective in identifying bias-conflicting samples. Their results suggest that earlier layers provide more fine-grained information to the reference classifier to identify and weigh samples from the worst-off groups. On the other hand, Tiwari et al. (2024) uses an ensemble of biased classifiers as the reference classifier. They upweight samples based on the proportion of reference classifiers that incorrectly classify them. Intuitively, samples misclassified by most reference classifiers in the ensemble are more likely to be from the worst-case group, thus providing a more reliable weighting scheme.

Another line of work identifies worst-case groups using the gradient magnitude during the optimization or the confidence of the reference classifier. Specifically, Ahn et al. (2022) hypothesized that minority samples get higher gradients when training a model with the generalized cross-entropy loss (Zhang & Sabuncu, 2018). Based on this hypothesis, the authors propose a method that involves two steps to mitigate bias. In the first step, a *biased* model is trained using the generalized cross-entropy loss to amplify the gradient of minority samples. In the second step, the final classifier is trained using data points sampled with probability proportional to their gradient provided by the reference model. Samples are, therefore, weighted proportionally to their gradient magnitude in the first step. Zhao et al. (2023) follows a similar approach but updates the biased model in the first step only using samples whose predictions are highly confident (mostly samples from the majority group). In the second step, sample weights are derived using the prediction probability of the biased auxiliary model such that samples with low confidence in prediction (mostly from the minority group) will receive higher weights. This weighting scheme enforces the alignment of gradients of the different groups during the training to mitigate bias.

**Fairness for free.** There is a line of work that shows that some existing deep learning techniques inherently have a positive impact on the worst-case group or some fairness metrics such as equal opportunity (Chai et al., 2022; Idrissi et al., 2022; Ko et al., 2023). For instance, Chai et al. (2022) theoretically demonstrated the connection between label smoothing and reweighing, i.e., with smooth labels, the model focuses more on samples that are hard to classify. The authors then empirically demonstrated the utility of label smoothing via knowledge distillation to improve fairness without relying on demographic information during the training. The framework consists of teacher-student models, where the teacher with a larger capacity is trained to overfit the training data, and its logits are used as smooth labels to train the student model. Remarkably,

the authors show that knowledge distillation not only improves the worst-case groups but can also effectively improve fairness in terms of equalized odds. On the other hand, Idrissi et al. (2022) show that applying simple data balancing techniques such as subsampling or reweighting can improve the test accuracy of the worst-case group and demonstrate that simple class balancing can be used when the group information is not available. While Kenfack et al. (2021b); Ko et al. (2023) show that ensemble models positively impact fairness, particularly for minority groups.

### 6.2.3 Invariant Representation Learning

Invariant representation learning is a group of methods aiming to learn representations invariant to different environments or domains. In contrast to robust optimization that optimizes for distributions close to the empirical distribution, invariant representation learning is a more general framework aiming to optimize for distributions that are eventually far from the empirical distribution (Arjovsky et al., 2019). A representation $\Phi(x)$ is environment invariant if, for any given environment $e_1$ and $e_2$ from the environment space $\mathcal{E}^{obs}$, it satisfies:

$$\mathbb{E}\left[y|\Phi(x) = h, e_1\right] = \mathbb{E}\left[y|\Phi(x) = h, e_2\right]$$
$$\forall h \in \mathcal{H}, \forall e_1, e_2 \in \mathcal{E}^{obs} \tag{14}$$

In other words, a representation satisfying this relation induces the same conditional class labels across environments. This formulation has different applications for privacy, domain adaptation, and fairness(Zhao et al., 2022). For fairness, the invariant representation formulation (Equation 14) is similar to the equation 1 for statistical parity, where demographic groups replace the environments. When group information is available, the invariant representation can be achieved through Invariant Risk Representation (IRM) (Arjovsky et al., 2019) or fair representation learning (Zemel et al., 2013; Madras et al., 2018; Kenfack et al., 2023a).

When group information is unavailable, Creager et al. (2021) proposes a method to automatically infer the environment (or group) from data. The proposed method infers group partition using a fixed classifier trained to maximize the invariance violation using a soft group assignment function ($q(e|x, y)$) optimized over the training data. Once the group assignment is acquired, existing IRM or GroupDro methods can enforce a representation that satisfies Equation 14.

### 6.2.4 Evaluation Protocol in the Rawlsian Fairness Setting.

In this setting, the model is trained without access to sensitive attributes. However, the evaluation of the performances of the worst-case groups is performed using partitions of the datasets based on sensitive attributes assumed known for the evaluation. Demographic groups are generally defined using an intersectionality of the sensitive attributes. For instance, Lahoti et al. (2020) and Chai et al. (2022) establish demographic groups in the Adult and Compas datasets based on a combination of *Race* and *Sex*. They report groups' worst-case performance (accuracy or AUC) using an intersection of sensitive attribute values, such as [White, Male], [White, Female], [Black, Female], and [Black, Male].

As shown in Table 5, Rawlsian fairness methods presented in this section mostly focus on improving fairness in terms of worst-case group performance. In particular, none of the methods presented target statistical parity. Moreover, experimental results show that methods optimizing for the worst-case performance fail to improve the statistical parity. On the other hand, results from some work showed improving the worst-case performance can also benefit group fairness metrics such as equalized odds and disparate impact are also considered (Lahoti et al., 2020; Chai et al., 2022).

### 6.3 Enforcing fairness Under Noisy Demographic Information

As discussed in Section 3.2.2, sensitive attributes could be noisy because they have been corrupted during data collection. Noise could also be introduced when the sensitive attributes are estimated or collected with privacy-preserving mechanisms. Recent works have emerged to mitigate bias w.r.t the true sensitive attribute when only a noisy version of sensitive attributes is available, i.e., there is a limited number of clean sensitive attributes. A straightforward method involves naïvely enforcing fairness using the noisy protected attributes, expecting to enhance fairness for the true protected groups. The empirical evidence presented by Gupta et al.

(2018) suggests that this objective can be realized when the noise in the protected attribute's space arises from estimation, i.e., pseudo-sensitive attributes are derived through an attribute predictor model. However, the model's bias could still be higher than that of a model with fairness on the true sensitive attributes (Celis et al., 2021; Gupta et al., 2018; Kallus et al., 2022).

Several works attempt to provide theoretical bounds on fairness violation when the sensitive attribute is noisy or design noise tolerant and noise robust approaches.

### 6.3.1 Bounding Fairness Violation.

When fairness constraints are applied using noisy protected attributes, one is interested in finding the conditions or assumptions that guarantee fairness improvement on the true protected groups. In this regard, Awasthi et al. (2020) theoretically analyze the impact of noisy attributes on the post-processing technique proposed by Hardt et al. (2016) (see Section 4.2) to improve fairness in terms of equalized odds. The post-processing mechanism takes a trained classifier and adjusts its decision boundary to satisfy fairness constraints while maintaining the accuracy as much as possible (Hardt et al., 2016). The authors show that when a certain conditional assumption is satisfied ($\hat{Y} \perp \hat{A} | (A, Y)$) and the sum of the corruption probabilities of each sensitive attribute value is upper bounded by one (i.e., $\sum_a P(\hat{A} \neq A | Y = y, A = a) \leq 1, \ \forall a \in \{0, 1\}, y \in \{0, 1\}$), the post-processing fairness mechanism using the noisy protected attribute $\hat{A}$ can be robust to the noise in the attribute space, i.e., mitigate bias w.r.t the true protected groups. This means that constraints imposed on the noisy protected groups do not exacerbate fairness violations in the true protected groups. However, this fairness violation bound by Awasthi et al. (2020) is only validated on a post-processing technique. On the other hand, Wang et al. (2020) provide an upper bound on the fairness violation for in-processing constraint optimization techniques. They demonstrate that if the total variation of conditional probability distributions of the data, given the true and noisy protected groups, is bounded for all groups, then fairness criteria will be satisfied w.r.t the true groups within the specified total variation bounds for each group. In practice, the corruption rate of the sensitive attributes can be used as the upper bound of the total variation between the conditional distributions, i.e., $P(A \neq \hat{A} | A = j) \leq \lambda, \ \forall j \in A$. The authors show that this bound on the fairness violation holds for various fairness metrics using performance disparity rates, such as demographic parity and equal opportunities. Pursuing a comparable objective, Bharti et al. (2022) establishes a worst-case upper bound on fairness violation in the context of equalized odds. This upper bound is defined under the assumption that the attribute classifier is sufficiently accurate and makes fewer misclassifications than the label classifier. However, this assumption may only hold true for a limited number of applications.

While evaluating the fairness violation bounds of the naïve approach that employs noisy protected attributes could be valuable for practitioners, these assessments often fail to provide fairness guarantees for the true protected groups. Another line of research concentrates on developing methods beyond the naive utilization of noisy protected attributes. Instead, these methods address the noise to construct fair models that exhibit tolerance to variations in sensitive attributes, ultimately providing guarantees of fairness improvement for the true protected groups. We broadly classify these methods into the Noise Tolerance and Noise Correction methods.

### 6.3.2 Noise Tolerant Methods.

We have seen in section 6.2.1 that DRO can control the worst-case error of groups with certain sizes when demographic information is unavailable. The optimization process is defined around the distance metric over probability distributions and the neighboring distributions' maximum distance (radius) likely to include all groups Hashimoto et al. (2018). Access to noisy protected attributes can help define a more meaningful boundary that improves fairness on true protected groups. In this spirit, Wang et al. (2020) formulate DRO using the corruption rate of the sensitive attributes as the upper bound of the divergence metric in the robust constraints. The corruption rate is assumed to be an upper bound of the total variation between the conditional probability distributions of the data given the true and noisy protected groups. The authors show this formulation is guaranteed to improve fairness in the true groups. However, it can lead to higher overall classification errors due to the assumption about the corruption rate in the robust loss. Wang et al. (2020) propose an alternative approach that uses *soft group assignment*. Soft group assignment leverages an axillary

dataset, where the true sensitive attributes are available, to compute sample weights. Each weight is an estimated probability that the sample belongs to a given demographic group. The proposed method builds upon Kallus et al.'s (2022) definition of robust fairness criteria for efficient bias assessment to design a robust optimization approach that is guaranteed to improve the fairness of the true groups while maintaining better accuracy compared to the DRO base method. In (Lamy et al., 2019), the authors propose a noise-tolerant approach that utilizes noise estimation methods, such as a mutually contaminated distribution, to model noise in the sensitive attribute space. The noise model provides corruption proportions for the sensitive attributes. The study demonstrates that any fairness-enhancing technique, accepting a minimum fairness violation parameter $\tau$, can be made robust to noise. This is achieved by deriving a new fairness violation from the noise model, i.e., $\tau' = (1 - \alpha - \beta) \cdot \tau$, where $\alpha$ and $\beta$ represent the (estimated) noise rates for each demographic group. In particular, the authors illustrate the ability to maintain fairness in clean sensitive attributes when running the exponentiated gradient fairness mechanism by Agarwal et al., with the noise scaled $\tau'$ instead of $\tau$ as the fairness violation tolerance.

### 6.3.3   Noise Correction Methods.

Chen et al. (2022a) propose an approach that involves correcting noisy sensitive attributes collected with privacy preservation techniques, such as differential privacy. This method assumes the availability of clean and private sensitive attributes, which train a model to "correct" the noisy sensitive attributes. The authors use the Corruption Matrix proposed by Hendrycks et al. (2018) to perform the correction. This is a $K \times K$ matrix that models the corruption process of sensitive attributes and contains probabilities that a label $i$ is corrupted to class $j$. The matrix is obtained by training a model on the noisy part of the sensitive attribute. Using the corrected sensitive attributes, Chen et al. (2022a) adapted the *adversarial debaising* (presented in Section 4.2) algorithm to enforce fairness w.r.t statistical parity (equation 1).

## 6.4   Enforcing Fairness Under Privacy-Preserving Demographics

One way to alleviate privacy restrictions on sensitive attributes is the design of fairness-enhancing methods employing mechanisms to preserve the privacy of the sensitive attribute. These mechanisms, offering strong privacy guarantees, can effectively address privacy concerns on access to demographic information. The most popular mechanisms include *trusted third party, secure multiparty computation, and differential privacy* .

### 6.4.1   Trusted Third Party

A trusted third party can be used to provide access to private demographic information. In this setup, the third party can hold sensitive information private and allow data centers or models to access it while providing a privacy guarantee to users. Sometimes, the third party might not provide access to the sensitive attributes. Still, it can only perform fairness analysis, e.g., given the classifier outcomes, the third returns its group fairness score. Hu et al. (2019) assumed a trusted third party with sensitive attributes to adapt different fairness algorithms to enforce fairness via the trusted third party. The trusted party is used to assess fairness violations by providing a signal to select the best model in terms of fairness. Hu et al. (2021) assume different trusted agents hold sensitive attributes and aim at training a fair model on non-private data without directly exchanging demographic information with the trusted agents. However, such approaches are vulnerable to inferential attacks (Hu et al., 2021; Ferry et al., 2023), i.e., attacks to reconstruct sensitive information. In addition, it is difficult in practice to find a trusted party that will collect or possess useful demographic information for a variety of tasks and datasets. This could be alleviated using methods that ensure strong privacy when the server or the third party cannot be trusted (Lowy & Razaviyayn, 2021; Lowy et al., 2023a). Under these methods, individuals can share their sensitive information even if they do not trust the third party.

### 6.4.2   Secure Multiparty Computation (SMC).

Originating in the 1980s, protocols for Secure Multiparty Computation aim to compute a function based on inputs from multiple parties in a distributed manner while ensuring that only the computation result is disclosed and the inputs of each party remain confidential. In an ideal scenario, a reliable third party

would handle the entire computation process, delivering the result to all participants before erasing any transaction memory. SMC, however, aims to facilitate this process without requiring such a third party. Yao first formulated what is now known as Yao's Millionaires problem, providing a provably secure solution for a two-party comparison problem Yao (1986). Since Yao's groundbreaking work, the SMC field has seen numerous innovations. One such advancement is `Blind Justice`, a framework introduced by Kilbertus et al. (2018). This framework enables the creation of fair machine-learning models without requiring access to sensitive attributes. Specifically, `Blind Justice` first employs additive secret sharing Shamir (1979) to randomly distribute parts of the sensitive attribute's value between two non-colluding entities: the model creator and the regulator. These two parties then engage in a two-server secure protocol Mohassel & Zhang (2017) to develop a fair machine-learning model. Although SMC helps maintain the sensitive attribute's confidentiality during the training phase, it is still possible to deduce this attribute from the fair model Jagielski et al. (2019); Ferry et al. (2023).

### 6.4.3 Differential Privacy

Several approaches have been proposed to train fair models with differential privacy (presented in Section 4.3) guarantees w.r.t to the sensitive attributes (Mozannar et al., 2020; Jagielski et al., 2019; Tran et al., 2022; Lowy et al., 2023b; Tran et al., 2021b;a). These methods generally involve transforming existing non-DP fair algorithms into fair DP algorithms using random noise. In particular, Lamy et al. (2019) show that one can release $(\epsilon, 0)-$differentially private sensitive attributes by randomly flipping the sensitive attribute with a probability $p = \frac{1}{\exp(\epsilon)+1}$ (Local DP). As previously discussed, imposing fairness constraints on noisy attributes does not guarantee fairness improvement for the true groups. To overcome this, Mozannar et al. (2020) propose a two-step method using existing fairness algorithms to train fair models with fairness and privacy guarantees on the true protected group. The proposed method initiates by applying the exponentiated gradient (Agarwal et al., 2018) to enforce fairness constraints on private (noisy) attributes. To ensure fairness guarantees on the true groups, Mozannar et al. adopt the post-processing technique introduced by Hardt et al. in the second step. This adaptation helps to derive a predictor that ensures fairness improvement on true attributes. Given the utilization of randomized response in collecting protected attributes, the fairness constraints in the optimization process are defined using the conditional probability of the true protected attribute given the private one to account for the noise and ensure no fairness violations on true groups. Similarly, Jagielski et al. (2019) integrate DP in the postprocessing of Hardt et al. by adding Laplace noise in the computed conditional probabilities before solving the resulting linear problem to drive the fair classifier. However, this postprocessing technique generally incurs a higher drop in accuracy and requires sensitive attributes at test time. To overcome this, Jagielski et al. (2019) incorporate DP in the in-processing fairness algorithm by Agarwal et al. (2018) (exponentiated gradient). The training is made DP by adding a Laplace noise to the gradient of the Auditor model during the two-player zero-sum game employed in the original algorithm to optimize the fairness constraints (Agarwal et al., 2018). On the other hand, Lowy et al. (2023b) argue that previous methods do not provide fairness and privacy convergence guarantees when the training is done using mini-batch stochastic gradient descent as in most large-scale deep learning models. Lowy et al. use regularization technique by Lowy et al. (2021) to enforce fairness constraints in ERM . This regularized loss is optimized using a DP variant of the stochastic gradient descent-ascent of Lin et al. (2020). In their method, Gaussian noise is added to the gradient of the weights to ensure differential privacy during training. Similarly, the work of Tran et al. (2021a) proposes a training inspired from *DP-Stochastic Gradient Descent* (DP-SGD) (Abadi et al., 2016) for training deep learning models with privacy guarantees. In DP-SGD, the gradient of each sample is clipped, and noise is added before updating the model's weights. Tran et al. use a Lagrangian formulation of ERM with fairness constraints. In addition to the Gaussian noise added to the model's weights, Gaussian noise is added to the updates of the Lagrangian multipliers to ensure differential privacy w.r.t sensitive attributes.

On the other hand, Tran et al. (2022) leverage the PATE framework (Papernot et al., 2016) to propose two methods to build fair models with strong privacy guarantees for demographic information. In the first approach, the teachers are trained to predict demographic information, and the student is trained with fairness constraints on the noisy aggregated demographic information from teachers. In the second approach, each teacher is trained to predict the target labels with fairness constraints w.r.t to true (private) protected attributes. The goal is to transfer the fairness of the teachers to the student model. The authors argue

that both approaches effectively bound and enhance fairness on true protected groups while simultaneously providing privacy guarantees for sensitive attributes.

Ensuring the privacy of the sensitive attribute in fair learning, however, introduces a tradeoff between fairness and privacy, i.e., smaller privacy budgets yield might models with higher bias (Lowy et al., 2021; Tran et al., 2021a). This leads to the triple accuracy-fairness-privacy tradeoff dilemma. Moreover, differentially private methods such as PATE and DP-SDG have shown disparate impacts on the model accuracy, i.e., the drop in accuracy is higher for minority groups (Uniyal et al., 2021). This suggests that training fair models under these frameworks might exacerbate unfairness in terms of Rawslian Max-Min fairness, i.e., worsen performance for the minority groups.

### 6.4.4 Evaluation Protocol in Noisy and Private Demographic Setting.

When evaluating methods in noisy sensitive attribute setups, a noise model is considered for corrupting sensitive attributes. For example, Lamy et al. (2019); Awasthi et al. (2020); Wang et al. (2020); Celis et al. (2021) consider a stochastic matrix $H \in [0,1]^{p \times p}$, i.e., $\sum_{j \in [p]} H_{ij} = 1$, where $H_{ij}$ is the corruption probability. This noise model is applied to the training data. The testing dataset is assumed clean for evaluation. In other settings, the noise model is defined based on a randomized response of sensitive attributes, i.e., a local differential private mechanism was applied to the sensitive attribute to ensure privacy. The evaluation protocol in privacy-preserving settings does not differ greatly from settings where demographic information is available. Specifically, the sensitive attributes are privately used during the training and testing phase (Tran et al., 2021b; Hu et al., 2019; Mozannar et al., 2020). The primary goal is to construct a fair model with strong privacy guarantees concerning the demographic data used in training. In ablation studies, existing methods introduce variations in the corruption probability of sensitive attributes or the privacy budget to assess their impact on the proposed methods. For instance, Chen et al. (2022b) demonstrate that, in the context of local differential privacy, reducing the privacy budget — resulting in a higher corruption rate in the sensitive attribute space — degrades fairness performance.

Fairness-enhancing methods under noisy and private demographic settings can improve group fairness metrics such as statistical parity , equalized odds , and equal opportunity . This is possible thanks to access to some demographic information; although private or noisy, existing methods aim at effectively controlling fairness on true protected attributes.

## 7 Auditing Bias Under Missing Protected Attributes

While auditing bias in AI systems generally requires access to protected attributes, regulations and laws prohibiting their use raise the need to design alternative methods for efficient bias assessment. We Existing methods attempt to provide efficient bias estimation of the true attributes when proxy attributes are available (Baines & Courchane, 2014; Chen et al., 2019; Kallus et al., 2022; Awasthi et al., 2021; Kallus et al., 2022), bias assessment under privacy-preserving of the sensitive attribute (Park et al., 2022a; Toreini et al., 2023), and bias assessment using group-free metrics (Liu et al., 2023).

### 7.1 Auditing Bias Using Proxy Protected Attributes.

When estimating biases w.r.t true demographic groups using proxy demographic information, the proxy model can exhibit biases/errors in predicting the true sensitive features, leading to adverse effects in disparities assessment on downstream tasks. For example, Baines & Courchane (2014) show that bias assessment using an approximation of racial information tends to overestimate the true disparity in a mortgage dataset. In general, for probabilistic models, the estimation of the sensitive attribute is *thresholded*, meaning that group membership is assigned when the output probability of the classifier is greater than a defined threshold $q \in [0.5, 1)$:

$$\hat{a}_i = \begin{cases} 0, \ \mathbb{P}\left(a_i = 0 \mid x_i\right) < q, \\ 1, \ \mathbb{P}\left(a_i = 1 \mid x_i\right) > q, \\ \text{NA, otherwise} \end{cases} \tag{15}$$

With this estimation, some samples (NA) are not used to compute the disparity when the proxy model outcome is lower than the threshold. Considering the equation 1, the demography disparity ($\hat{\Delta}$) of a model that uses a *thresholded estimator* of the sensitive attribute can be measured as follows:

$$\hat{\Delta} = \hat{\mu}(0) - \hat{\mu}(1)$$
$$\text{with } \hat{\mu}(\alpha) = \frac{\sum_N^{i=1} \mathbb{I}(\hat{a}_i = \alpha \mid x_i)\hat{y}_i}{\sum_N^{i=1} \mathbb{I}(\hat{a}_i = \alpha \mid x_i)}, \ \alpha \in \{0,1\} \tag{16}$$

where $\mathbb{I}$ represents the indicator function and $\hat{y}_i = f(x_i)$ the predicted label using the classifier $f$. The overestimation or the underestimation of the true disparity is defined by comparing the computed disparity based on the thresholded estimator ($\hat{\Delta}$) with disparities obtained from the true sensitive feature ($\Delta$), i.e., $\hat{\Delta} - \Delta$. This difference's positive and negative values represent the overestimation and underestimation of the true disparity, respectively. As protected attribute classifiers always provide noisy labels when estimating unknown protected features, algorithms for auditing unfairness should account for the noise in sensitive attributes for better bias estimation or mitigation. In particular, Chen et al. (2019) perform an analysis of *thresholded* proxy models and show that bias assessment with the derived proxies can result in an overestimation or underestimation of the true disparities, i.e., the disparity in terms of the positive prediction rate computed using the proxy-protected attribute is relatively lower or greater than the positive prediction rate w.r.t the true protected groups. The authors suggest that instead of computing the disparities based on thresholded estimation of the sensitive features, using a soft group assignment within a Weighted Estimator (WE) is more efficient. i.e., by replacing $\hat{\mu}$ in the equation 16 with the following $\hat{\mu}_w$:

$$\hat{\mu}_w(\alpha) = \frac{\sum_N^{i=1} \mathbb{P}(a_i = \alpha \mid x_i)\hat{y}_i}{\sum_N^{i=1} \mathbb{P}(a_i = \alpha \mid x_i)}, \ \alpha \in \{0,1\} \tag{17}$$

With the weighted estimator, the uncertainty of the probabilistic proxy model is propagated into the final estimation of the demographic parity, which may be more useful for the outcome disparity evaluations when proxy models are used (Chen et al., 2019). Contrastingly, Kallus et al. (2022) demonstrate that without observing the joint distribution $P(A, Y, X)$, it is challenging to precisely identify the true disparity measure when relying on proxy attributes derived from marginals, specifically $P(X, Y)$ and $P(A, X)$. This limitation arises due to the insufficient information provided by the corresponding marginal distributions, preventing the unique determination of the joint distribution. Consequently, numerous valid full joint distributions can provide various potential disparities, leading to ambiguity in the true disparity measure. This ambiguity can be alleviated only if an independence assumption is satisfied (i.e., $Y, \hat{Y} \perp A|X$) or there are few samples drawn from the joint distribution $P(A, Y)$ available. To reduce the set of possible disparity measures emerging from the marginal distributions, the Kallus et al. assume one has access to two datasets drawn from the marginals $P(A, X)$ and $P(\hat{Y}, Y, X)$. This assumption aligns with the scenario we outlined in Section 6.1.2, where the sensitive attribute is solely available in an auxiliary dataset. With the two datasets, the authors show that it is possible to limit the number of disparity values and to derive the closed-form of the disparity estimation for various fairness metrics such as equalized odds and equal opportunity.

Perhaps counterintuitively, Awasthi et al. (2021) demonstrate that the accuracy of the protected attribute classifier does not necessarily correlate with the accuracy of the bias estimation, i.e., the gap between the disparity measured using the true sensitive attributes and the predicted ones. Furthermore, the authors show that accounting for the uncertainty of the sensitive attribute predictor using an *active sampling* technique can yield a better estimation of the model disparity. Estimation is done using samples for which the sensitive attribute predictor is most certain. A limitation of using the probabilistic proxy models is the assumption that the features that correlate with the unknown sensitive features are known and available.

On the other hand, Fabris et al. (2023) formulate bias estimation without sensitive attributes as a *quantification* problem. The goal of quantification methods is to estimate the prevalence of a class label in a given sample, i.e., the proportion of the class in the given sample. A naive quantification method is the *classify and count* (CC) approach. It consists of training a classifier and counting the frequency of each class in the unlabelled dataset. Fabris et al. (2023) show that group fairness metrics can be formulated using prevalence scores.

For example, the equal opportunity metric (Equation 4) can be expressed using the prevalence score of the sensitive attribute as follows:

$$
\begin{aligned}
\mathrm{P}(\hat{Y} = 1 \mid A = a, Y = 1) &= \frac{\mathrm{P}(Y = 1, \hat{Y} = 1, A = a)}{\mathrm{P}(Y = 1, A = a)} \\
&= \underbrace{\frac{\mathrm{P}(A = a \mid Y = 1, \hat{Y} = 1)}{\mathrm{P}(A = a \mid Y = 1)}}_{\text{obtained from prevalence estimator}} \cdot \frac{\mathrm{P}(Y = 1, \hat{Y} = 1)}{\mathrm{P}(Y = 1)}
\end{aligned}
\tag{18}
$$

$$\forall a \in \{0, 1\}$$

This suggests that an efficient prevalence estimator can better estimate the true bias in a model. In particular, when applied to bias estimation using a proxy attribute classifier, the naive prevalence estimator with hard labels corresponds to the thresholded proxy model (Equation 16), and the estimator with soft labels corresponds to the weighted estimator presented above (Equation 17). However, the naive quantification approach is not robust to feature or label space distribution shifts. Fabris et al. (2023) leverage more principled existing quantification methods robust to distribution shift. For example, Bella et al. (2010) propose *adjusted classify and count* (ACC), a better prevalence estimator that accounts for true positive rate and false positive rate. On the unlabelled data, the true positive rates and false positive rates are estimated from the training data using K-fold cross-validation. (Fabris et al., 2023) provide empirical evidence that quantification-based approaches can be more effective in estimating the true bias in the dataset with missing sensitive attributes, compared to other methods such as thresholded proxy and weighted estimation,

Cornacchia et al. (2023) introduce CFlips (Counterfactual Flips), a new metric for auditing bias based on counterfactual reasoning. Given a trained model to be audited, a sample $x$ and its protected attribute $x_a$, when the model predicts a negative outcome for $x$ (i.e., denied a loan), a *counterfactual generator* is used to generate a set of samples that are closer to $x$ but for which the model predicts a positive prediction. The sensitive attribute classifier is then used to predict the sensitive attributes of samples in the counterfactual set of $x$. The bias score (CFlips score) is then calculated as the proportion of samples in the counterfactual set having a predicted sensitive attribute different from $x_a$. The intuition is that a model that does not encode any dependency to the sensitive attribute should have the same predicted sensitive attribute for all the counterfactual samples, i.e., a CFlips score equals one. Empirical results show that a model trained with fairness constraints tends to provide a CFlips score closer to one. In contrast, models without fairness constraints get a score closer to 0, meaning there is a majority sample in the counterfactual set with positive predictions assigned to a different demographic group. While these results show the efficiency of CFlips score for bias assessment, its effectiveness highly depends on the quality of the counterfactual generator.

While the bias estimation methods presented above can effectively assess the true fairness violation, they pose ethical or privacy risks due to proxies or inference of sensitive information about individuals using non-sensitive information.

## 7.2 Auditing Bias Under Privacy Preserving Protected Attributes.

Bias assessment under secured and privacy-preserving frameworks represents an alternative approach to using proxy-sensitive attributes. This group of methods generally considers a black box access to the model to be audited. For example, Park et al. (2022a) propose a Trusted Execution Environment (TEE) for secure and privacy-preserving bias assessment. This method leverages confidential computing technology, utilizing specialized hardware to store sensitive data in encrypted form and ensuring verifiability guarantees for computation integrity. Within the TEE, the proposed approach offers fairness certification and verification protocol for regulatory bodies and extends public verifiability of fairness certificates to end-users. However, this protocol is sensitive to attacks such as data poisoning for fairness (Solans et al., 2020). In such attacks, poisoning points can be crafted and introduced in the test set used by the regulator to mislead and enforce the issuing of a fairness certificate to a biased model. Toreini et al. (2023) propose a different protocol incorporating three key components: the black-box model under scrutiny, a group of individuals serving as auditors, and a dedicated *fairness computation module*. Auditors interact with the model through secured

and encrypted communications, receiving predictions in encrypted form. Subsequently, each auditor encrypts essential information for bias assessment—namely, the predicted outcome, ground truth label, and locally observed and privately held demographic information. The encrypted information is then independently transmitted to the fairness computation module, which conducts fairness assessments based on a given fairness notion. This evaluation protocol prioritizes the security and privacy of data by employing computation over encrypted data and integrating *zero knowledge proof* (Goldreich & Oren, 1994) for verifying the integrity of the computations. In addition, the proposed bias assessment protocol provides public access to users interested in verifying the bias auditing process.

### 7.3 Auditing Bias Without Group Labels.

Liu et al. (2023) introduce *Group-Free Group Fairness* a bias assessment process that does not rely on demographic group labels. This group fairness measure is based on *homophily*, a property in social networks where people sharing the same attributes (both sensitive and non-sensitive) are more likely to be connected than dissimilar people. In social science and social networks analysis of homophily, there is strong empirical evidence of homophily in demographic dimensions, i.e., in community networks, there is high connectivity between individuals sharing attributes such as race, gender, religion, and age (Verbrugge, 1977). In contrast to bias estimation methods using proxy group labels, *group-free group fairness* measures group disparities using pairwise similarities between individuals (Liu et al., 2023). The authors evaluate this approach's effectiveness in quantifying a model's disparities on several tasks, such as node classification, recommender systems, and information access maximization in a graph. The main drawback of group-free bias estimation comes from the assumption that there is a social network that exhibits the homophily property for all the demographic attributes of interest.

## 8 Conclusion and Future Directions

In this paper, we explored the issue of bias estimation and mitigation when the assumption of full access to demographic information does not hold. We presented various real-world scenarios where the large body of work on fairness using demographic information is not directly applicable. In particular, we presented the settings where the sensitive attributes are entirely missing, partially available, noisy, or available only under privacy guarantees. We presented existing works aiming at addressing fairness issues in each of these settings and provided a taxonomy. In particular, when demographic information is not available: 1) fairness can be enforced by focusing on improving the performance of the worst-case group. This goal can be achieved using methods from robust optimization, invariant representation learning, or reweighting using various methods of identifying protected subgroups; 2) related features or attribute classifiers can be used as a proxy to true demographic groups; 3) sensitive attributes can be noisy due data corruption (e.g., for privacy concerns), existing works focus on bounding fairness violation on the true protected groups, and designing methods that correct or are tolerant to noise in the attributes space; 4) fair and privacy-preserving models can be used to alleviate the privacy restrictions from regulators or laws. Despite this large and growing body of work, there are challenges to be addressed, given room for the following future perspectives:

**Bridging Fairness and Generalization.** In the survey, we highlighted the similarities between generalization and fairness, in particular for the Rawslsian Max-Min fairness. We observed that improving the performance of the model over the worst-case group can be formulated as a generalization problem, where different domains are considered as the unknown demographic groups. However, the fields of fairness and generalization are mostly studied independently. The work by Creager et al. (2021) makes initial steps in studying them jointly, by leveraging techniques from fairness without demographics to design a domain generalization method without domain knowledge. More broadly, some existing works aim to improve generalization by mitigating dataset bias, framed as spurious correlation (Nam et al., 2020; Zhao et al., 2023; Ahn et al., 2022). The main objective in mitigating spurious correlation in the data is to identify the *bias-conflicting* and *bias-aligned* samples (Nam et al., 2020), which are samples from the minority and the majority groups respectively. Besides their similarities, methods for generalization problems are often evaluated only on generalization benchmarks with known spurious correlations such as Waterbird, CMNIST,

and CIFAR100 datasets. It will be interesting if different proposed methods are also evaluated on fairness benchmarks to assess their effectiveness in improving the performance of worst-case demographic groups.

**Robustness to distribution shifts.** Most existing work presented in the survey assumes the data-generating process is fixed over time and does not consider dynamic settings where the model is deployed in changing environments (distribution shift). There are various reasons for the change in the environment such as changes in the covariate, label, or protected attribute distribution (Barrainkua et al., 2023; Prost et al., 2021). There are many practical scenarios in which distribution shifts occur, reflecting the constant evolution of our society, such as in healthcare (Chen et al., 2021; Finlayson et al., 2021). It is, therefore, essential to design robust, fair models without demographic information while preserving fairness guarantees when data-generating processes change in the deployed environment.

**Consistent Evaluation Protocols with Provable Fairness Guarantees.** When demographic information is missing, most existing approaches assume sensitive information is observed during the evaluation phase. The reasons for the unavailability of the sensitive attributes during the training phase might also apply to the testing phase or during internal bias screening. This makes the application of the proposed methods less realistic in real-world scenarios, where the sensitive attributes are also not available for bias assessment. Therefore, it is important to design bias estimation methods that make the same assumptions about the sensitive attributes across the training, validation, and testing stages. For example, when the sensitive attribute is noisy, the bias assessment on the testing set should also be performed on a noisy attribute set. This assessment should be done using bias assessment methods with provable fairness guarantees on true fairness violations, that also account for the noisy sensitive attributes. The same observation applies to fairness-enhancing methods that use proxy sensitive attributes where the test is made on true sensitive attributes instead of using bias assessment methods — over the proxy attributes — with provable fairness guarantees on the true sensitive attributes (Diana et al., 2022; Jung et al., 2022; Kenfack et al., 2023a). We have covered some of these methods in section 7, however, they are yet to be employed for evaluation on different benchmarks.

**Establishsing the Limits of Fairness with Missing Protected Attributes.** We observed throughout the survey that alternative solutions in enforcing fairness in missing protected attribute settings generally come with new dilemmas that are yet to be addressed. For example:

- Using proxy models to infer sensitive attributes poses privacy risks (Andrus et al., 2021). In fact, the prediction of sensitive information could also violate the privacy constraints from regulations that prohibit their collection and the restriction could also be applied to predicted sensitive information. Furthermore, predicting sensitive attributes can be unlawful and raises ethical concerns, e.g., inferring gender using a photo or inferring racial/ethnicity using last names. Besides, there is no way to evaluate the accuracy of the proxy-sensitive information when real values are not observed. This leads to a risk of incorrect bias estimation and discrimination exacerbation.

- Training fair models under privacy preservation of sensitive attributes raises the fairness-privacy tradeoffs dilemma (Tran et al., 2021a; Jagielski et al., 2019). We observed that privacy-preserving mechanisms such as differential privacy can provide strong privacy guarantees and alleviate the restrictions from regulators or laws. However, existing works show that the stronger the privacy of sensitive attributes the higher the fairness violation. While regulators or laws enforce both discrimination-free and privacy-preserving decision-making processes, it remains unclear which value between fairness and privacy should be prioritized when there is no tradeoff satisfying the different expectations. Moreover, privacy-preserving mechanisms exhibit disparate impact on minority groups (Bagdasaryan et al., 2019; Uniyal et al., 2021); their use in fair learning can worsen the performance in terms of worst-case groups.

- Enforcing worst-case group fairness does not necessarily mitigate the disparities between demographic groups (Chai et al., 2022; Ozdayi et al., 2021; Lahoti et al., 2020). We observed that while achieving the Rawlsian principle of distributed justice can be an effective fairness metric in different scenarios, it also inherits the critiques of the Rawlsian principles. In particular, the difficulty in defining and

targeting the *right* disadvantaged groups (Franke, 2021). On the other hand, while empirical evidence shows that aiming to improve the worst-case group can positively impact equalized odds, it generally fell short in improving group fairness compared to sensitive-attribute aware fair learning. These shortcomings, therefore, limit the practical application of Rawlsian Max-Min fairness in contexts where group fairness notions are required, particularly group notions in trade-offs with accuracy.

These dilemmas raise the need to clarify the limits of what can be achieved when the sensitive attributes—the most important information for bias mitigation and assessment—are missing. This would provide regulators and practitioners with tools and information to take appropriate actions for bias-free automated decision-making. Furthermore, addressing these dilemmas might require rethinking the formulation of unfairness in machine learning.

**Benchmarking Existing Methods.** A research direction that is worth exploring is to perform an empirical comparison of the existing methods in order to establish state-of-the-art performances on different fairness benchmarks. This will be particularly useful in providing the research community with methods that should be used for comparison, the evaluation process (choice of the worst-case group), and the evaluation metrics. In particular, we observed that some newly proposed methods to improve group fairness metrics, such as demographic parity, are compared with methods designed to improve worst-case performance. The benchmark will guide the community toward comparing methods on fairness metrics that they are designed to improve.

### Acknowledgments

SEK is supported by CIFAR and NSERC DG (2021-4086) and UA by NSERC DG (2022-04006).

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
