# OpenReview forum: "A Survey on Fairness Without Demographics"
_TMLR — Accepted by TMLR_

### Review · Reviewer_6Ysp · 2024-03-08

**Summary Of Contributions:**

This survey reviews the recent studies on fair machine learning without complete sensitive information. The survey considers several constraints on the sensitive attributes, including missing sensitive attributes, partial sensitive attributes, noisy sensitive attributes, and private sensitive attributes. Then, the survey introduces the fairness notions without complete sensitive information, including proxy fairness, fairness through unawareness, individual fairness, and Rawlsian fairness. Finally, the survey discusses the techniques for ensuring fairness, including enforcing fairness using proxy sensitive attributes, achieving Rawlsian max-min fairness, enforcing fairness under noisy sensitive information, and enforcing fairness under privacy-preserving demographics.

**Audience:**

Yes

**Claims And Evidence:**

Yes

**Requested Changes:**

The authors may further justify the rationale behind the current organization. Alternatively, the authors may consider reorganizing the techniques in a clearer way, i.e., consider how to reduce the overlap between different categories.

**Strengths And Weaknesses:**

Strength:
The survey provides a broad overview of the topic of fair machine learning without complete sensitive information. It covers a wide range of related works, and also delves into the concert concepts, methods, and findings of some important studies. It also evaluates the strengths, weaknesses, and contributions of different approaches.

Weaknesses:
In Page 2, the sentence “non-sensitive proxy features are generally unknown” is not accurate, since non-sensitive proxy features could be identified according to their correlation with the sensitive features.

The survey identifies several constraints on the sensitive features. I have two concerns related to this. (1) It seems that these constraints are conceptually overlapped. For example, the private sensitive attribute could be overlapped with the missing sensitive attribute, as in both cases the sensitive information may not be available to the decision maker. (2) The constraints are identified based on the reasons for incomplete sensitive information. However, from the technical perspective, we may not care why the sensitive information is missing. What we care about may be different types of incomplete information, such as completely missing information, partially missing information, noisy information, etc.

The first sentence in Section 3.3: “Unfairness or discrimination in a decision-making process made by humans is quite clear: it occurs when the outcome of a decision systematically depends on an individual’s protected attribute and not on characteristics that are useful in assessing that individual’s abilities with respect to the task or desired outcome.” is not accurate. As also discussed in the survey, even if the decision does not directly rely on the sensitive attribute, the decision could still be unfair due to the proxy features.
Also in Section 3.3, it is not meaningful to discuss the intention of an ML system.

In Section 4.1, Disparate Impact is just one way to measure Statistical Parity so it should not be listed as a separate fairness notion.

In Section 5.1, I don’t think individual fairness is irrelevant to sensitive attributes. This is when you define the distance/similarity metric, you need to ensure that the metric does not consider sensitive information.

Missing E in equation 13.

---

> ### Author Response · Authors · 2024-04-21
> **Reply to Reviewer 6Ysp**
>
> We thank the reviewer for the detailed feedback and comments. In the revised manuscript, we have highlighted all the changes requested in `violet` and as outlined below:
>
> > In Page 2, the sentence “non-sensitive proxy features are generally unknown” is not accurate, since non-sensitive proxy features could be identified according to their correlation with the sensitive features
>
>
> We referred to settings where sensitive features are not observed or used, thereby hindering the possibility of computing their correlation with non-sensitive ones. We have updated the statement to reflect that.
>
>
> > (1) It seems that these constraints are conceptually overlapped. For example, the private sensitive attribute could be overlapped with the missing sensitive attribute, as in both cases the sensitive information may not be available to the decision maker. (2) The constraints are identified based on the reasons for incomplete sensitive information. However, from the technical perspective, we may not care why the sensitive information is missing. What we care about may be different types of incomplete information, such as completely missing information, partially missing information, noisy information, etc.
>
> Our categorization is not based on the reason why the sensitive information is missing or incomplete, which might seem to overlap from this perspective.  We provide the reason for incomplete sensitive information to emphasize real-world situations where this might happen. Indeed, our categorization is based on existing technical contributions to enforce fairness, which differ in the assumptions and type of sensitive information available under each category we identified.
> While sensitive information might not be available to the decision maker in private and missing sensitive attribute setups, the two setups are different when model developers design fairness intervention.
> In particular, we observed that methods under private attribute setups generally assume some access to sensitive attributes and involve some privacy-preserving mechanisms. On the other hand, in the missing sensitive attribute setup, algorithms are designed to improve the performance of unknown minority groups implicitly and without privacy preservation concerns.
> Although the missing sensitive attribute category might result from privacy concerns, methods in this setup do not involve privacy-preserving techniques. This justifies why we group them in a category different from the private sensitive attribute setup.
>
>
> We have updated the manuscript to better highlight the motivation for this choice of categorization, which is based on existing technical contributions and technical challenges in enforcing fairness under each category.
>
>
> > In Section 4.1, Disparate Impact is just one way to measure Statistical Parity so it should not be listed as a separate fairness notion.
>
> We thank the reviewer for bringing this up. We have moved the definition of Disparate Impact (DI) under Statistical Parity. We clarified that DI is similar to Statistical Parity but uses the ratio instead of the difference.
>
>
> > In Section 5.1, I don’t think individual fairness is irrelevant to sensitive attributes. This is when you define the distance/similarity metric, you need to ensure that the metric does not consider sensitive information.
>
>
> The distance metric in individual fairness might depend on the sensitive attribute. However, such a distance metric would not be a good one since it uses irrelevant information to compare individuals’ ability to perform a task. For example, in the resume scoring, suppose two applicants $x$ and $y$ are very similar: they have similar years of experience, graduated from the same school, and have mastered the same programming languages. But $y$ is a woman; the task-specific similarity metric should ignore this fact since gender is irrelevant to determining who should be hired. Thus, individual fairness should not involve demographic information since protected information is irrelevant when comparing people’s abilities in decision-making processes.
>
> We thank the reviewer again for their time and feedback. We hope our changes address your concerns, and we are happy to provide further clarifications.

---

### Review · Reviewer_muA1 · 2024-04-08

**Summary Of Contributions:**

This submission is intended to be a survey on machine learning fairness, specifically concerning the setting where "Demographic" information is missing. In this case "Demographic" information refers to information that is considered "sensitive". For instance, this could be class information. This can obviously cause problems when considering class/group-based fairness concepts.

This paper provides a background on the problem of machine learning fairness, discusses first notions of fairness, followed by approaches for fairness.

**Audience:**

Yes

**Broader Impact Concerns:**

Fairness is an important topic to discuss broader impact concerns, but since this is a survey, I don't see any immediate issues.

**Claims And Evidence:**

Yes

**Requested Changes:**

* I suggest you re-write the introduction based on my comments in above.

* (throughout) “fair machine community”, "fair ML community", etc., be more consistent.

* (Table 1) What does “fairly” mean, Overall, the table (spacing) looks unprofessional, please fix.

* (Table 4) Looks slightly more professional, but still poorly formatted. Please make all tables look consistent.

* (Eqn. 4) The equation is incorrect.

* (Fig 1 and 2) Remove or revamp Fig 1 and 2.

* (throughout) General proofreading is needed. Though I don't notice a huge amount of technical errors, phrasing is often awkward.

* Section 6 is too verbose, but also lacks detail, please revise based on my points above.

**Strengths And Weaknesses:**

*Strengths*

* I like the way this paper categorizes the existing fairness methods.
* Sections 4 and 5 is the best part of the paper, succinct and informative.

*Weaknesses*

* In my opinion the introduction is cluttered and off-putting. The introduction of a survey paper should explain the reason why the paper is being written. What is the missing context in the literature? What is the information you plan to present? How will you present it? This kind of introduction that you have written would be better left for a background section.

* Tables are inconsistently formatted and hard to read

* Figure 2 is strange. Each sub-heading should be clearly be a technique. I understand that each of these words are associated with techniques, but it is confusing

* This paper is quite verbose, yet still fails to go deep enough into the details of the algorithms for it to be useful.

* Section 6 is too verbose, but simultaneously lacking detail. Overall, I feel that this section fails as a survey. Perhaps it would be better to focus less on summarizing a long list of other papers, and more to explain the overarching concepts in each of the "types" of approaches you have identified, explaining the pros and cons of each with respect to the metrics that were identified in Section 5.

---

> ### Author Response · Authors · 2024-04-21
> **Reply to Reviewer muA1**
>
> We thank the reviewer for the detailed feedback and insightful suggestions. We have carefully addressed all the points raised in the revised manuscript in `red` and as outlined below.
>
>
> > I suggest you re-write the introduction based on my comments in above.
>
>
> We welcome and appreciate the reviewer’s suggestion. In the revised manuscript, we have modified the introduction to address the concerns and restructured as suggested. Specifically, we emphasized the context of the paper, the topics covered, their motivations, and how they are presented.
>
> > (throughout) “fair machine community”, "fair ML community", etc., be more consistent.
>
> We fixed the inconsistencies throughout the paper.
>
> > (Table 1) What does “fairly” mean, Overall, the table (spacing) looks unprofessional, please fix.
>
> By *fairly*, we meant the survey briefly covers fairness without demographic information. We have fixed the formatting issues with the table and clarified the confusion.
>
> > Section 6 is too verbose, but also lacks detail, please revise based on my points above.
>
> We have revised section 6 to make it more succinct. While we acknowledge that not all papers are covered in detail, and we provided enough information for most papers to help the reader grasp the core idea.  Moreover, section 6 presents an overview of each method we identified under each category, along with their pros and cons. We also emphasize how each technique addresses the challenges identified in each category and which fairness metrics they can improve.
>
> Thank you again for your time and valuable feedback, which significantly helped to improve the paper. We hope the changes have addressed your concerns, and we will be happy to provide more clarifications where needed.

---

### Review · Reviewer_HWuh · 2024-04-10

**Summary Of Contributions:**

This submission provides a survey of research on algorithmic fairness without access to sensitive demographic data. An overview of related surveys is given, followed by a discussion of fair ML settings in which protected attributes might be (partially) missing or noisy. Then, background is given and different notions of fairness without demographics are discussed. Finally, an overview of techniques for fair ML without demographics is given and then the conclusion discusses some directions for future work.

**Audience:**

Yes

**Claims And Evidence:**

Yes

**Requested Changes:**

Please see above in **Weaknesses**. I believe that all of these changes would strengthen the work. I think that at least addressing the majority of them would be critical to securing my recommendation.

**Strengths And Weaknesses:**

**Strengths:**

-Fair learning without demographic data is an important research topic and a survey on this topic will be useful to the community.

-The survey is fairly extensive and comprehensive (no pun intended).

**Weaknesses:**

-Writing and grammar could use work. There are many run-on sentences (e..g. the penultimate sentence of the second paragraph of section 4.3) and run-on paragraphs. Also, there are some sloppy punctuation/grammar/spelling errors that I encourage the authors to find by carefully re-reading the paper word for word.

-Table 1: "Fairly" in column 3 is a bit unclear

-Please elaborate on the essential differences between the submission and the apparently closely related work of Ashurt & Weller (2023).

-Citation styles are sloppy and inconsistent: e.g., you should often be using parenthetical citations, such as (Dwork et al., 2006) instead of Dwork et al. (2006), for in-sentence citations.

-Section 4.1 seems to be missing accuracy parity

-Section 4.2 seems to be missing in-processing and pre-processing techniques

-Discussion of the meaning of Definition 1 is missing. For example, what does DP w.r.t. sensitive attributes guarantee and what does it not guarantee? For example, if sensitive attributes are independent of non-sensitive attributes, then it guarantees that no adversary can learn much more about any individual than they could learned had that individual's data never been used. On the other hand, such a guarantee does not hold if sensitive attributes are strongly correlated with the non-sensitive attributes.

-Is the paragraph on PATE necessary to include in the paper? It seems to be a bit of a non sequitur.

-It would be nice to have the order of the boxes in Figure 1 aligned with the order of the subsections in which these boxes are discussed.

-Def. 2: Clarify that individual fairness is a property of M and that (D, d) is fixed. Also, it would be nice to elaborate on the resume scoring example by identifying suitable D, d, and M in this example, to further clarify Def. 2.

-Can the notation in Eq. 7 be aligned with the notation in Def. 2?

-First paragraph in section 5.3: This does not seem to apply exclusively to Rawlsian fairness. Can't this be said of the other fairness notions described in Figure 1? If yes, then this paragraph seems to belong before Section 5.1. If no, then Figure 1 seems misleading.

-Since Definition 3 involves the sensitive demographic information a, it is unclear how this is a notion of fairness without demographic data. Please clarify.

-"In contrast to other group-based fairness notions..." : Equalized odds/opportunity also allow for disparate outcomes.

-Section 6: One important thing that is missing from this section is discussion of which different techniques can cover which fairness notions (e.g. demographic parity, equalized odds, accuracy parity, etc.). For example, Rawlsian techniques seem to apply specifically to accuracy parity, whereas other techniques can provide different fairness notions (e.g. Lowy et al., 2023 can handle equalized odds and demographic parity).

-"The core advantage of federated learning is to enhance the privacy..." : it should be noted that local storage is not sufficient to prevent data from being leaked. It would be worth mentioning works on differentially private federated learning without a trusted server, e.g. [1] and [2] cited below. These works are also relevant to the discussion in section 6.4.1 about trusted third party approaches being vulnerable to inference attacks.

-"These methods generally involve....the randomized response mechanism." is not really an accurate sentence. Several of the works on DP fair ML that you cited in the previous sentence do not use randomized response.

[1] Lowy, Andrew, and Meisam Razaviyayn. "Private Federated Learning Without a Trusted Server: Optimal Algorithms for Convex Losses." The Eleventh International Conference on Learning Representations. 2023.

[2] Lowy, Andrew, Ali Ghafelebashi, and Meisam Razaviyayn. "Private non-convex federated learning without a trusted server." International Conference on Artificial Intelligence and Statistics. PMLR, 2023.

---

> ### Author Response · Authors · 2024-04-21
> **Reply to Reviewer HWuh**
>
> We thank the reviewer for taking the time to review our manuscript and providing valuable feedback. In the revised manuscript, we have highlighted all the changes requested in  `blue` and addressed the comments as outlined below.
>
> > Writing and grammar could use work. There are many run-on sentences (e..g. the penultimate sentence of the second paragraph of section 4.3) and run-on paragraphs. Also, there are some sloppy punctuation/grammar/spelling errors that I encourage the authors to find by carefully re-reading the paper word for word.
>
>
> We have thoroughly proofread and fixed typos and inconsistencies in the text. Many sentences have also been reformulated to ease the flow.
>
> > Table 1: "Fairly" in column 3 is a bit unclear
>
> The term “Fairly” in Table 1, means the survey discusses a few methods aiming to achieve fairness without demographics.
>
> > Please elaborate on the essential differences between the submission and the apparently closely related work of Ashurt & Weller (2023).
>
>
> Ashurt & Weller (2023) discuss challenges and techniques to collect demographic data, methods providing protections for collection, and alternative methods such as sensitive information inference. They also presented methods for fair learning without demographic data. In contrast, our work differs on many points:
> We conceptualized the constraints in sensitive attributes.  Our work identifies and discusses five main constraints or assumptions over sensitive information used in existing fairness-enhancing techniques without demographic information.
> We presented and discussed fairness notions that do not rely on sensitive information.
> We proposed a taxonomy of fairness enhancing technique without demographics using the constraints we identified. Under each category in the taxonomy, we presented a systematic review of existing techniques, their pros and cons, and the fairness metric they can handle.
> We also highlighted the limitations, current challenges, and open research questions in  bias mitigation without demographic data.
> We highlighted the differences in the related work section of the paper.
>  Citation styles are sloppy and inconsistent: e.g., you should often be using parenthetical citations, such as (Dwork et al., 2006) instead of Dwork et al. (2006), for in-sentence citations.
>
> We have identified and fixed all inconsistencies in citations.
>
> > Section 4.1 seems to be missing accuracy parity
>
> We have added accuracy parity in Section 4.1.
> > Section 4.2 seems to be missing in-processing and pre-processing techniques.
>
> Fairness-enhancing techniques presented in Section 4.2 are methods adapted by fairness techniques to enforce fairness without demographic information. We only aimed to provide a general description of these algorithms used in subsequent works presented in Section 6.
>
> > Since Definition 3 involves the sensitive demographic information a, it is unclear how this is a notion of fairness without demographic data. Please clarify.
>
>
> Although the definition itself includes demographic information $a$, we emphasized that $a$ is unknown. The demographic information is used just to formalize the definition. The Rawlsian notion of fairness is defined around unknown least advantaged groups, which can be any group of a certain size. Groups are not predefined, unlike in group fairness notions.
>
> > Is the paragraph on PATE necessary to include in the paper? It seems to be a bit of a non sequitur.
>
> As PATE is used in only one work presented in section 6.1, its presence in the paper is indeed optional; we have removed it.
>
> > It would be nice to have the order of the boxes in Figure 1 aligned with the order of the subsections in which these boxes are discussed.
>
> We updated Figure 1 in the revised manuscript.
>
> > Def. 2: Clarify that individual fairness is a property of M and that (D, d) is fixed. Also, it would be nice to elaborate on the resume scoring example by identifying suitable D, d, and M in this example, to further clarify Def. 2.
>
>
> We elaborated on the resume scoring example: Suppose two applicants $x$ and $y$ are very similar: they have similar years of experience, graduated from the same school, and have mastered the same programming languages. However, $y$ is a woman; the similarity metric should ignore this fact since gender is irrelevant to determining who should be hired. Thus, the distance between $x$ and $y$ should be small, e.g., $d(x,y)=.01$. If the model $M$ assigns the probabilities .9 and .7 to $x$ and $y$ respectively, the distance between their scores $D(M(x), M(y))=.2$, assuming $D$ is defined as the statistical distance. which is higher than the distance between individuals. Therefore, $M$ fails to satisfy individual fairness since the Lipschitz mapping in Eq 2 is not satisfied, suggesting similar applicants are not treated similarly.

---

> > ### Author Response · Authors · 2024-04-21
> > **Reply to Reviewer HWuh (Continued)**
> >
> > > It would be nice to have the order of the boxes in Figure 1 aligned with the order of the subsections in which these boxes are discussed.
> >
> > We have updated Figure 1 accordingly.
> >
> > > Can the notation in Eq. 7 be aligned with the notation in Def. 2?
> >
> >
> > We fixed the inconsistencies in the notation.
> >
> > > First paragraph in section 5.3: This does not seem to apply exclusively to Rawlsian fairness. Can't this be said of the other fairness notions described in Figure 1? If yes, then this paragraph seems to belong before Section 5.1. If no, then Figure 1 seems misleading.
> >
> >
> > As correctly suggested by the reviewer, we removed this paragraph as it doesn’t belong to section 5.3.
> >
> > > -"In contrast to other group-based fairness notions..." : Equalized odds/opportunity also allow for disparate outcomes.
> >
> > Thank you for the comment. We have revised the sentence as “In contrast to other group-based fairness notions, Rawlsian fairness does not aim to mitigate the disparities in a given metric across groups but to make the worst-performing group as good as possible”. More specifically, given a set of models, applying Rawlsian Min-Max fairness requires selecting the best-performing model on the worst-off groups. In contrast, other group-based fairness metrics aim to equalize performances across a given metric.
> >
> >
> > > Section 6: One important thing that is missing from this section is discussion of which different techniques can cover which fairness notions (e.g. demographic parity, equalized odds, accuracy parity, etc.). For example, Rawlsian techniques seem to apply specifically to accuracy parity, whereas other techniques can provide different fairness notions (e.g. Lowy et al., 2023 can handle equalized odds and demographic parity).
> >
> >
> > Tables 4 & 5 provide evaluation metrics of methods in each category. Furthermore, we have added a discussion in the evaluation protocol of each sub-section about the fairness metrics that each group of methods can handle. More specifically, we highlighted that Rawlsian techniques focus on improving the worst-case performance, and some methods can also improve equalized odds or disparate impact. Methods under private or noisy protected attributes can handle group fairness metrics since they have access to some sort of demographic information that is noisy or restricted with privacy guarantees. This also applies to methods using proxy demographic information.
> >
> > > "The core advantage of federated learning is to enhance the privacy..." : it should be noted that local storage is not sufficient to prevent data from being leaked. It would be worth mentioning works on differentially private federated learning without a trusted server, e.g. [1] and [2] cited below. These works are also relevant to the discussion in section 6.4.1 about trusted third party approaches being vulnerable to inference attacks.
> >
> > We have updated the statement to reflect the risk of data leakage stored locally and added discussion of related works on differentially private learning without a trusted server.
> >
> > > "These methods generally involve....the randomized response mechanism." is not really an accurate sentence. Several of the works on DP fair ML that you cited in the previous sentence do not use randomized response.
> >
> > Thank you for the comment. We have updated the sentence to  “These methods generally involve transforming existing non-DP fair algorithms into fair DP algorithms using random noise.”
> >
> > We thank the reviewer again for their time and insightful comments. We hope the changes have addressed your concerns, and we will be happy to provide more clarifications where needed.

---

### Decision · Action_Editor_NDkt · 2024-05-28

**Recommendation:** Accept as is

**Comment:**

The reviewers are convinced by the paper and it provides general understanding and analysis for fairness without demographics.

**Audience:**

The general TMLR audience would be interested.

**Claims And Evidence:**

The claims made in the submission are supported by accurate, convincing and clear evidence.